# Pitfall of Optimism: Distributional Reinforcement Learning by Randomizing Risk Criterion

**Taehyun Cho**[1], **Seungyub Han**[1,3], **Heesoo Lee**[1], **Kyungjae Lee**[2], **Jungwoo Lee**[1]*

[1] Seoul National University, [2] Chung-Ang University, [3] Hodoo AI Labs

{talium, seungyubhan, algol7240, junglee}@snu.ac.kr
{kyungjae.lee}@ai.cau.ac.kr

## Abstract

Distributional reinforcement learning algorithms have attempted to utilize estimated uncertainty for exploration, such as optimism in the face of uncertainty. However, using the estimated variance for optimistic exploration may cause biased data collection and hinder convergence or performance. In this paper, we present a novel distributional reinforcement learning algorithm that selects actions by randomizing risk criterion to avoid one-sided tendency on risk. We provide a perturbed distributional Bellman optimality operator by distorting the risk measure and prove the convergence and optimality of the proposed method with the weaker contraction property. Our theoretical results support that the proposed method does not fall into biased exploration and is guaranteed to converge to an optimal return. Finally, we empirically show that our method outperforms other existing distribution-based algorithms in various environments including Atari 55 games.

## 1 Introduction

Distributional reinforcement learning (DistRL) learns the stochasticity of returns in the reinforcement learning environments and has shown remarkable performance in numerous benchmark tasks. DistRL agents model the approximated distribution of returns, where the mean value implies the conventional Q-value [2, 6, 13, 25] and provides more statistical information (e.g., mode, median, variance) for control. Precisely, DistRL aims to capture *intrinsic (aleatoric)* uncertainty which is an inherent and irreducible randomness in the environment. Such learned uncertainty gives rise to the notion of risk-sensitivity, and several distributional reinforcement learning algorithms distort the learned distribution to create a risk-averse or risk-seeking decision making [8, 12].

Despite the richness of risk-sensitive information from return distribution, only a few DistRL

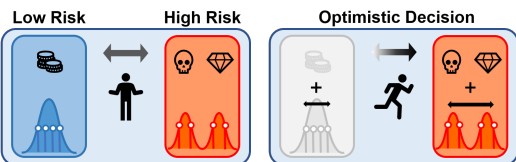

Figure 1: Illustrative example of why a biased risk criterion (naïve optimism) can degrade performance. Suppose two actions have similar expected returns, but different variances (intrinsic uncertainty). (**Left**) If an agent does not specify the risk criterion at the moment, the probability of selecting each action should be similar. (**Right**) As OFU principle encourages to decide uncertain behaviors, the empirical variance from quantiles was used as an estimate of uncertainty. [19, 21, 23]. However, optimistic decision based on empirical variance inevitably leads a risk-seeking behavior, which causes biased action selection.

methods [11, 21, 26, 35, 42] have tried to employ its benefits for exploration strategies which is essential in deep RL to find an optimal behavior within a few trials. The main reason is that the exploration strategies so far is based on *parametric (epistemic)* uncertainty which arise from insufficient or

---

*Corresponding author

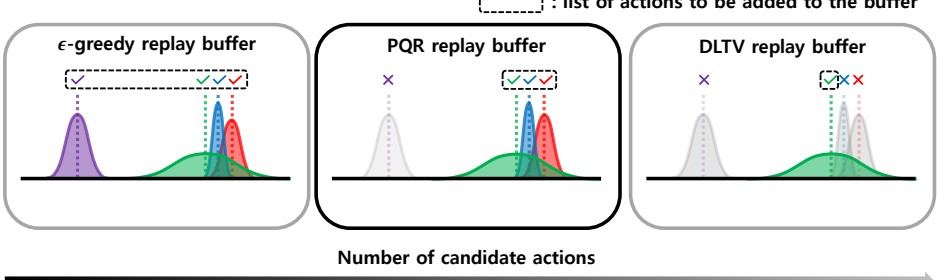

Figure 2: An illustrative example of proposed algorithm (PQR). Each distribution represents the empirical PDF of return. PQR benefits from excluding inferior actions and promoting unbiased selection with regards to high intrinsic uncertainty through randomized risk criterion.

inaccurate data. In particular, *Optimism in the face of uncertainty* (OFU) is one of the fundamental exploration principles that employs parametric uncertainty to promote exploring less understood behaviors and to construct confidence set. In bandit or tabular MDP settings, OFU-based algorithms select an action with the highest upper-confidence bound (UCB) of parametric uncertainty which can be considered as the optimistic decision at the moment [5, 10].

However, in deep RL, it is hard to trivially estimate the parametric uncertainty accurately due to the black-box nature of neural networks and high-dimensionality of state-action space. Without further computational task, the estimated variance from distribution is extracted as a mixture of two types of uncertainty, making it difficult to decompose either component. For example, DLTV [21] was proposed as a distribution-based OFU exploration that decays bonus rate to suppress the effect of intrinsic uncertainty, which unintentionally induces a risk-seeking policy. Although DLTV is the first attempt to introduce OFU in distRL, we found that consistent optimism on the uncertainty of the estimated distribution still leads to biased exploration. We will refer to this side-effect as *one-sided tendency on risk*, where selecting an action based on a fixed risk criterion degrades learning performance. In Section 4, we will demonstrate the one-sided tendency on risk through a toy experiment and show that our proposed randomized approach is effective to avoid this side effect.

In this paper, we introduce *Perturbed Distributional Bellman Optimality Operator (PDBOO)* to address the issue of biased exploration caused by a one-sided tendency on risk in action selection. We define the distributional perturbation on return distribution to re-evaluate the estimate of return by distorting the learned distribution with perturbation weight. To facilitate deep RL algortihm, we present *Perturbed Quantile Regression (PQR)* and test in Atari 55 games comparing with other distributional RL algorithms that have been verified for reproducibility by official platforms [4, 29].

In summary, our contributions are as follows.

- A randomized approach called perturbed quantile regression (PQR) is proposed without sacrificing the original (risk-neutral) optimality and improves over naïve optimistic strategies.

- A sufficient condition for convergence of the proposed Bellman operator is provided without satisfying the conventional contraction property.

## 2 Backgrounds & Related works

### 2.1 Distributional RL

We consider a Markov decision process (MDP) which is defined as a tuple $(\mathcal{S}, \mathcal{A}, P, R, \gamma)$ where $\mathcal{S}$ is a finite state space, $\mathcal{A}$ is a finite action space, $P : \mathcal{S} \times \mathcal{A} \times \mathcal{S} \to [0, 1]$ is the transition probability, $R$ is the random variable of rewards in $[-R_{\max}, R_{\max}]$, and $\gamma \in [0, 1)$ is the discount factor. We define a stochastic policy $\pi(\cdot|s)$ which is a conditional distribution over $\mathcal{A}$ given state $s$. For a fixed policy $\pi$, we denote $Z^\pi(s, a)$ as a random variable of return distribution of state-action pair $(s, a)$ following the policy $\pi$. We attain $Z^\pi(s, a) = \sum_{t=0}^{\infty} \gamma^t R(S_t, A_t)$, where $S_{t+1} \sim P(\cdot|S_t, A_t)$, $A_t \sim \pi(\cdot|S_t)$ and $S_0 = s$, $A_0 = a$. Then, we define an action-value function as $Q^\pi(s, a) = \mathbb{E}[Z^\pi(s, a)]$ in $[-V_{\max}, V_{\max}]$ where $V_{\max} = R_{\max}/(1 - \gamma)$. For regularity, we further notice that the space of

action-value distributions $\mathcal{Z}$ has the first moment bounded by $V_{\max}$:

$$\mathcal{Z} = \left\{ Z : \mathcal{S} \times \mathcal{A} \to \mathscr{P}(\mathbb{R}) \middle| \; \mathbb{E}[|Z(s,a)|] \leq V_{\max}, \forall (s,a) \right\}.$$

In distributional RL, the return distribution for the fixed $\pi$ can be computed via dynamic programming with the distributional Bellman operator defined as,

$$\mathcal{T}^\pi Z(s,a) \overset{D}{=} R(s,a) + \gamma Z(S', A'), \quad S' \sim P(\cdot|s,a), \; A' \sim \pi(\cdot|S')$$

where $\overset{D}{=}$ denotes that both random variables share the same probability distribution. We can compute the optimal return distribution by using the distributional Bellman optimality operator defined as,

$$\mathcal{T} Z(s,a) \overset{D}{=} R(s,a) + \gamma Z(S', a^*), \quad S' \sim P(\cdot|s,a), \; a^* = \underset{a'}{\arg\max} \; \mathbb{E}_Z[Z(S', a')].$$

Bellemare et al. [2] have shown that $\mathcal{T}^\pi$ is a contraction in a maximal form of the Wasserstein metric but $\mathcal{T}$ is not a contraction in any metric. Combining with the expectation operator, $\mathbb{E}\mathcal{T}$ is a contraction so that we can guarantee that the expectation of $Z$ converges to the optimal state-action value. Another notable difference is that the convergence of a return distribution is not generally guaranteed to be unique, unless there is a total ordering $\prec$ on the set of greedy policies.

## 2.2 Exploration on Distributional RL

To combine with deep RL, a parametric distribution $Z_\theta$ is used to learn a return distribution. Dabney et al. [13] have employed a quantile regression to approximate the full distribution by letting $Z_\theta(s,a) = \frac{1}{N} \sum_{i=1}^{N} \delta_{\theta_i(s,a)}$ where $\theta$ represents the locations of a mixture of $N$ Dirac delta functions. Each $\theta_i$ represents the value where the cumulative probability is $\tau_i = \frac{i}{N}$. By using the quantile representation with the distributional Bellman optimality operator, the problem can be formulated as a minimization problem as,

$$\theta = \arg\min_{\theta'} D\left(Z_{\theta'}(s_t, a_t), \mathcal{T} Z_{\theta^-}(s_t, a_t)\right) = \arg\min_{\theta'} \sum_{i,j=1}^{N} \frac{\rho_{\hat{\tau}_i}^\kappa \left(r_t + \gamma \theta_j^-(s_{t+1}, a') - \theta_i'(s_t, a_t)\right)}{N}$$

where $(s_t, a_t, r_t, s_{t+1})$ is a given transition pair, $\hat{\tau}_i = \frac{\tau_{i-1} + \tau_i}{2}$, $a' := \arg\max_{a'} \; \mathbb{E}_Z[Z_\theta(s_{t+1}, a')]$, $\rho_{\hat{\tau}_i}^\kappa(x) := |\hat{\tau}_i - \delta_{\{x<0\}}| \mathcal{L}_\kappa(x)$, and $\mathcal{L}_\kappa(x) := x^2/2$ for $|x| \leq \kappa$ and $\mathcal{L}_\kappa(x) := \kappa(|x| - \frac{1}{2}\kappa)$, otherwise.

Based on the quantile regression, Dabney et al. [13] have proposed a quantile regression deep Q network (QR-DQN) that shows better empirical performance than the categorical approach [2], since the quantile regression does not restrict the bounds for return.

As deep RL typically did, QR-DQN adjusts $\epsilon$-greedy schedule, which selects the greedy action with probability $1 - \epsilon$ and otherwise selects random available actions uniformly. The majority of QR-DQN variants [12, 38] rely on the same exploration method. However, such approaches do not put aside inferior actions from the selection list and thus suffers from a loss [28]. Hence, designing a schedule to select a statistically plausible action is crucial for efficient exploration.

In recent studies, Mavrin et al. [21] modifies the criterion of action selection for efficient exploration based on optimism in the face of uncertainty. Using left truncated variance as a bonus term and decaying ratio $c_t$ to suppress the intrinsic uncertainty, DLTV was proposed as an uncertainty-based exploration in DistRL without using $\epsilon$-greedy schedule. The criterion of DLTV is described as:

$$a^* = \underset{a'}{\arg\max} \left( \mathbb{E}_P[Z(s', a')] + c_t \sqrt{\sigma_+^2(s', a')} \right), \quad c_t = c\sqrt{\frac{\log t}{t}}, \; \sigma_+^2 = \frac{1}{2N} \sum_{i=\frac{N}{2}}^{N} (\theta_{\frac{N}{2}} - \theta_i)^2,$$

where $\theta_i$'s are the values of quantile level $\tau_i$.

## 2.3 Risk in Distributional RL

Instead of an expected value, risk-sensitive RL is to maximize a pre-defined risk measure such as Mean-Variance [41], Value-at-Risk (VaR) [9], or Conditional Value-at-Risk (CVaR) [30, 31] and results in different classes of optimal policy. Especially, Dabney et al. [12] interprets risk measures

as the expected utility function of the return, i.e., $\mathbb{E}_Z[U(Z(s,a))]$. If the utility function $U$ is linear, the policy obtained under such risk measure is called *risk-neutral*. If $U$ is concave or convex, the resulting policy is termed as *risk-averse* or *risk-seeking*, respectively. In general, a *distortion risk measure* is a generalized expression of risk measure which is generated from the distortion function.

**Definition 2.1.** Let $h : [0,1] \to [0,1]$ be a **distortion function** such that $h(0) = 0, h(1) = 1$ and non-decreasing. Given a probability space $(\Omega, \mathcal{F}, \mathbb{P})$ and a random variable $Z : \Omega \to \mathbb{R}$, a **distortion risk measure** $\rho_h$ corresponding to a distortion function $h$ is defined by:

$$\rho_h(Z) := \mathbb{E}^{h(\mathbb{P})}[Z] = \int_{-\infty}^{\infty} z \frac{\partial}{\partial z}(h \circ F_Z)(z) dz,$$

where $F_Z$ is the cumulative distribution function of $Z$.

In fact, non-decreasing property of $h$ makes it possible to distort the distribution of $Z$ while satisfying the fundamental property of CDF. Note that the concavity or the convexity of distortion function also implies risk-averse or seeking behavior, respectively. Dhaene et al. [14] showed that any distorted expectation can be expressed as weighted averages of quantiles. In other words, generating a distortion risk measure is equivalent to choosing a reweighting distribution.

Fortunately, DistRL has a suitable configuration for risk-sensitive decision making by using distortion risk measure. Chow et al. [8] and Stanko and Macek [34] considered risk-sensitive RL with a CVaR objective for robust decision making. Dabney et al. [12] expanded the class of policies on arbitrary distortion risk measures and investigated the effects of a distinct distortion risk measures by changing the sampling distribution for quantile targets $\tau$. Zhang and Yao [40] have suggested QUOTA which derives different policies corresponding to different risk levels and considers them as options. Moskovitz et al. [24] have proposed TOP-TD3, an ensemble technique of distributional critics that balances between optimism and pessimism for continuous control.

## 3 Perturbation in Distributional RL

### 3.1 Perturbed Distributional Bellman Optimality Operator

To choose statistically plausible actions which may be maximal for certain risk criterion, we will generate a distortion risk measure involved in a pre-defined constraint set, called an *ambiguity set*. The ambiguity set, originated from distributionally robust optimization (DRO) literature, is a family of distribution characterized by a certain statistical distance such as *$\phi$-divergence* or *Wasserstein distance* [15, 32]. In this paper, we will examine the ambiguity set defined by the discrepancy between distortion risk measure and expectation. We say the sampled reweighting distribution $\xi$ as *(distributional) perturbation* and define it as follows:

**Definition 3.1.** (Perturbation Gap, Ambiguity Set) Given a probability space $(\Omega, \mathcal{F}, \mathbb{P})$, let $X : \Omega \to \mathbb{R}$ be a random variable and $\Xi = \left\{\xi : 0 \le \xi(w) < \infty, \int_{w \in \Omega} \xi(w)\mathbb{P}(dw) = 1\right\}$ be a set of probability density functions. For a given constraint set $\mathcal{U} \subset \Xi$, we say $\xi \in \mathcal{U}$ as a **(distributional) perturbation** from $\mathcal{U}$ and denote the $\xi$−weighted expectation of $X$ as follows:

$$\mathbb{E}_\xi[X] := \int_{w \in \Omega} X(w)\xi(w)\mathbb{P}(dw),$$

which can be interpreted as the expectation of $X$ under some probability measure $\mathbb{Q}$, where $\xi = d\mathbb{Q}/d\mathbb{P}$ is the Radon-Nikodym derivative of $\mathbb{Q}$ with respect to $\mathbb{P}$. We further define $d(X; \xi) = |\mathbb{E}[X] - \mathbb{E}_\xi[X]|$ as **perturbation gap** of $X$ with respect to $\xi$. Then, for a given constant $\Delta \ge 0$, the **ambiguity set** with the bound $\Delta$ is defined as

$$\mathcal{U}_\Delta(X) = \left\{\xi \in \Xi : d(X; \xi) \le \Delta\right\}.$$

For brevity, we omit the input $w$ from a random variable unless confusing. Since $\xi$ is a probability density function, $\mathbb{E}_\xi[X]$ is an induced risk measure with respect to a reference measure $\mathbb{P}$. Intuitively, $\xi(w)$ can be viewed as a distortion to generate a different probability measure and vary the risk tendency. The aspect of using distortion risk measures looks similar to IQN [12]. However, instead of changing the sampling distribution of quantile level $\tau$ implicitly, we reweight each quantile from the

ambiguity set. This allows us to control the maximum allowable distortion with bound $\Delta$, whereas the risk measure in IQN does not change throughout learning. In Section 3.3, we suggest a practical method to construct the ambiguity set.

Now, we characterize *perturbed distributional Bellman optimality operator* (PDBOO) $\mathcal{T}_\xi$ for a fixed perturbation $\xi \in \mathcal{U}_\Delta(Z)$ written as below:

$$\mathcal{T}_\xi Z(s,a) \overset{D}{=} R(s,a) + \gamma Z(S', a^*(\xi)), \quad S' \sim P(\cdot|s,a), \ a^*(\xi) = \underset{a'}{\operatorname{argmax}} \ \mathbb{E}_{\xi,P}[Z(s',a')].$$

Notice that $\xi \equiv 1$ corresponds to a base expectation, i.e., $\mathbb{E}_{\xi,P} = \mathbb{E}_P$, which recovers the standard distributional Bellman optimality operator $\mathcal{T}$. Specifically, PDBOO perturbs the estimated distribution only to select the optimal behavior, while the target is updated with the original (unperturbed) return distribution.

If we consider the time-varying bound of ambiguity set, scheduling $\Delta_t$ is a key ingredient to determine whether PDBOO will efficiently explore or converge. Intuitively, if an agent continues to sample the distortion risk measure from a fixed ambiguity set with a constant $\Delta$, there is a possibility of selecting sub-optimal actions after sufficient exploration, which may not guarantee eventual convergence. Hence, scheduling a constraint of ambiguity set properly at each action selection is crucial to guarantee convergence.

Based on the quantile model $Z_\theta$, our work can be summarized into two parts. First, we aim to minimize the expected discrepancy between $Z_\theta$ and $\mathcal{T}_\xi Z_{\theta^-}$ where $\xi$ is sampled from ambiguity set $\mathcal{U}_\Delta$. To clarify notation, we write $\mathbb{E}_\xi[\cdot]$ as a $\xi-$weighted expectation and $\mathbb{E}_{\xi \sim \mathscr{P}(\mathcal{U}_\Delta)}[\cdot]$ as an expectation with respect to $\xi$ which is sampled from $\mathcal{U}_\Delta$. Then, our goal is to minimize the perturbed distributional Bellman objective with sampling procedure $\mathscr{P}$:

$$\min_{\theta'} \mathbb{E}_{\xi_t \sim \mathscr{P}(\mathcal{U}_{\Delta_t})}[D(Z_{\theta'}(s,a), \mathcal{T}_{\xi_t} Z_{\theta^-}(s,a))] \tag{1}$$

where we use the Huber quantile loss as a discrepancy on $Z_{\theta'}$ and $\mathcal{T}_\xi Z_{\theta^-}$ at timestep $t$. In typical risk-sensitive RL or distributionally robust RL, the Bellman optimality equation is reformulated for a pre-defined risk measure [8, 33, 39]. In contrast, PDBOO has a significant distinction in that it performs dynamic programming that adheres to the risk-neutral optimal policy while randomizing the risk criterion at every step. By using min-expectation instead of min-max operator, we suggest unbiased exploration that can avoid leading to overly pessimistic policies. Furthermore, considering a sequence $\xi_t$ which converges to 1 in probability, we derive a sufficient condition of $\Delta_t$ that the expectation of any composition of the operators $\mathbb{E}\mathcal{T}_{\xi_{n:1}} := \mathbb{E}\mathcal{T}_{\xi_n}\mathcal{T}_{\xi_{n-1}} \cdots \mathcal{T}_{\xi_1}$ has the same unique fixed point as the standard. These results are remarkable that we can apply the diverse variations of distributional Bellman operators for learning.

## 3.2 Convergence of the perturbed distributional Bellman optimality operator

Unlike conventional convergence proofs, PDBOO is time-varying and not a contraction, so it covers a wider class of Bellman operators than before. Since the infinite composition of time-varying Bellman operators does not necessarily converge or have the same unique fixed point, we provide the sufficient condition in this section. We denote the iteration as $Z^{(n+1)} := \mathcal{T}_{\xi_{n+1}} Z^{(n)}$, $Z^{(0)} = Z$ for each timestep $n > 0$, and the intersection of ambiguity set as $\bar{\mathcal{U}}_{\Delta_n}(Z^{(n-1)}) := \bigcap_{s,a} \mathcal{U}_{\Delta_n}(Z^{(n-1)}(s,a))$.

**Assumption 3.2.** Suppose that $\sum_{n=1}^\infty \Delta_n < \infty$ and $\xi_n$ is uniformly bounded.

**Theorem 3.3.** *(Weaker Contraction Property) Let $\xi_n$ be sampled from $\bar{\mathcal{U}}_{\Delta_n}(Z^{(n-1)})$ for every iteration. If Assumption 3.2 holds, then the expectation of any composition of operators $\mathbb{E}\mathcal{T}_{\xi_{n:1}}$ converges, i.e., $\mathbb{E}\mathcal{T}_{\xi_{n:1}}[Z] \to \mathbb{E}[Z^*]$. Moreover, the following bound holds,*

$$\sup_{s,a} \left| \mathbb{E}[Z^{(n)}(s,a)] - \mathbb{E}[Z^*(s,a)] \right| \le \sum_{k=n}^\infty \left( 2\gamma^{k-1}V_{max} + 2\sum_{i=1}^k \gamma^i(\Delta_{k+2-i} + \Delta_{k+1-i}) \right).$$

Practically, satisfying Assumption 3.2 is not strict to characterize the landscape of scheduling. Theorem 3.3 states that even without satisfying $\gamma$-contraction property, we can show that $\mathbb{E}[Z^*]$ is the fixed point for the operator $\mathbb{E}\mathcal{T}_{\xi_{n:1}}$. However, $\mathbb{E}[Z^*]$ is not yet guaranteed to be "unique" fixed point for any $Z \in \mathcal{Z}$. Nevertheless, we can show that $\mathbb{E}[Z^*]$ is, in fact, the solution of the standard Bellman optimality equation, which is already known to have a unique solution.

**Algorithm 1** Perturbed QR-DQN (PQR)

---
**Input:** $(s, a, r, s'), \gamma \in [0, 1)$, timestep $t > 0$, $\epsilon > 0$, concentration $\boldsymbol{\beta}$

    Initialize $\Delta_0 > 0$.

    $\Delta_t \leftarrow \Delta_0 t^{-(1+\epsilon)}$.                                                      // Assumption 3.2

    $\boldsymbol{\xi} \leftarrow \max\left(\mathbf{1}^N + \Delta_t(N\boldsymbol{x} - \mathbf{1}^N), 0\right)$ where $\boldsymbol{x} \sim \mathrm{Dir}(\boldsymbol{\beta})$     // Sample $\xi \sim \bar{\mathcal{U}}_{\Delta_t}(Z^{(t)})$

    $\boldsymbol{\xi} \leftarrow N\boldsymbol{\xi}/\sum \xi_i$                                             // Refine as a weighting function

    $a^* \leftarrow \mathrm{argmax}_{a'} \, \mathbb{E}_{\boldsymbol{\xi}}[Z(s', a')]$                 // Select greedy action with perturbed return

    $\mathcal{T}\theta_j \leftarrow r + \gamma\theta_j(s', a^*), \quad \forall j$         // Target update with unperturbed distribution

    $t \leftarrow t + 1$

**Output:** $\sum_{i=1}^N \mathbb{E}_j[\rho_{\hat{\tau}_i}^\kappa(\mathcal{T}\theta_j - \theta_i(s, a))]$

---

**Theorem 3.4.** *If Assumption 3.2 holds, $\mathbb{E}[Z^*]$ is the unique fixed point of Bellman optimality equation for any $Z \in \mathcal{Z}$.*

As a result, PDBOO generally achieves the unique fixed point of the standard Bellman operator. Unlike previous distribution-based or risk-sensitive approaches, PDBOO has the theoretical compatibility to obtain a risk-neutral optimal policy even if the risk measure is randomly sampled during training procedure. For proof, see Appendix A.

### 3.3 Practical Algorithm with Distributional Perturbation

In this section, we propose a **perturbed quantile regression (PQR)** that is a practical algorithm for distributional reinforcement learning. Our quantile model is updated by minimizing the objective function (1) induced by PDBOO. Since we employ a quantile model, sampling a reweight function $\xi$ can be reduced into sampling an $N$-dimensional weight vector $\boldsymbol{\xi} := [\xi_1, \cdots, \xi_N]$ where $\sum_{i=1}^N \xi_i = N$ and $\xi_i \geq 0$ for all $i \in \{1, \cdots, N\}$. Based on the QR-DQN setup, note that the condition $\int_{w \in \Omega} \xi(w)\mathbb{P}(dw) = 1$ turns into $\sum_{i=1}^N \frac{1}{N}\xi_i = 1$, since the quantile level is set as $\tau_i = \frac{i}{N}$.

A key issue is how to construct an ambiguity set with bound $\Delta_t$ and then sample $\boldsymbol{\xi}$. A natural class of distribution for practical use is the *symmetric Dirichlet distribution* with concentration $\boldsymbol{\beta}$, which represents distribution over distributions. (i.e. $\boldsymbol{x} \sim \mathrm{Dir}(\boldsymbol{\beta})$.) We sample a random vector, $\boldsymbol{x} \sim \mathrm{Dir}(\boldsymbol{\beta})$, and define the reweight distribution as $\boldsymbol{\xi} := \mathbf{1}^N + \alpha(N\boldsymbol{x} - \mathbf{1}^N)$. From the construction of $\boldsymbol{\xi}$, we have $1 - \alpha \leq \xi_i \leq 1 + \alpha(N - 1)$ for all $i$ and it follows that $|1 - \xi_i| \leq \alpha(N - 1)$. By controlling $\alpha$, we can bound the deviation of $\xi_i$ from 1 and bound the perturbation gap as

$$\sup_{s,a} |\mathbb{E}[Z(s, a)] - \mathbb{E}_\xi[Z(s, a)]| = \sup_{s,a} \left| \int_{w \in \Omega} Z(w; s, a)(1 - \xi(w))\mathbb{P}(dw) \right|$$
$$\leq \sup_{w \in \Omega} |1 - \xi(w)| \sup_{s,a} \mathbb{E}[|Z(s, a)|] \leq \sup_{w \in \Omega} |1 - \xi(w)| V_{\max} \leq \alpha(N - 1)V_{\max}.$$

Hence, letting $\alpha \leq \frac{\Delta}{(N-1)V_{\max}}$ is sufficient to obtain $d(Z; \xi) \leq \Delta$ in the quantile setting. We set $\boldsymbol{\beta} = 0.05 \cdot \mathbf{1}^N$ to generate a constructive perturbation $\xi_n$ which gap is close to the bound $\Delta_n$. For Assumption 3.2, our default schedule is set as $\Delta_t = \Delta_0 t^{-(1+\epsilon)}$ where $\epsilon = 0.001$.

## 4 Experiments

Our experiments aim to investigate the following questions.

1. Does randomizing risk criterion successfully escape from the biased exploration in stochastic environments?

2. Can PQR accurately estimate a return distribution?

3. Can a perturbation-based exploration perform sucessfully as a behavior policy for the full Atari benckmark?

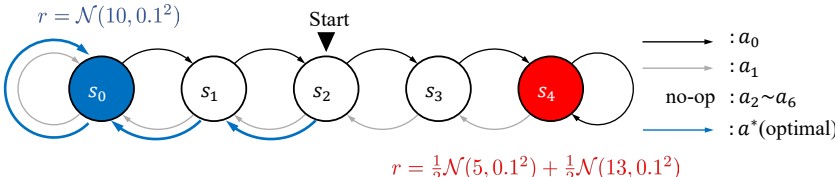

Figure 3: Illustration of the N-Chain environment [27] with high uncertainty starting from state $s_2$. To emphasize the intrinsic uncertainty, the reward of state $s_4$ was set as a mixture model composed of two Gaussian distributions. Blue arrows indicate the risk-neutral optimal policy in this MDPs.

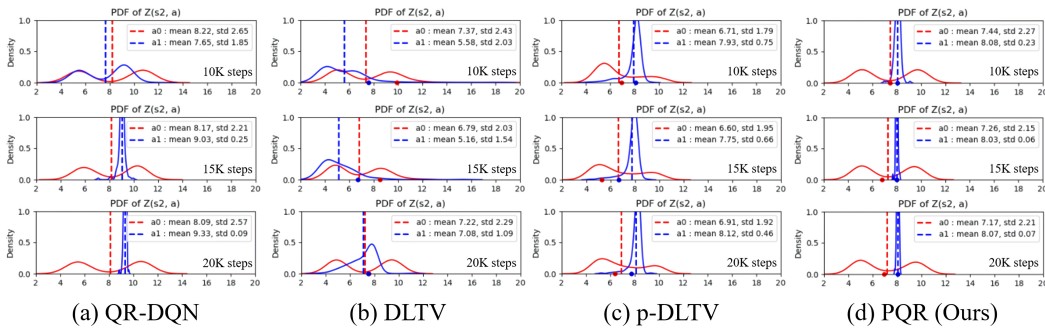

| (a) QR-DQN | (b) DLTV | (c) p-DLTV | (d) PQR (Ours) |

Figure 4: Empirical return distribution plot in N-Chain environment. The ground truth of each distribution is $\gamma^2 \mathcal{N}(10, 0.1^2)$ and $\gamma^2[\frac{1}{2}\mathcal{N}(5, 0.1^2) + \frac{1}{2}\mathcal{N}(13, 0.1^2)]$. Each dot represents an indicator for choosing action. Since QR-DQN does not depend on other criterion, the dots are omitted.

## 4.1 Learning on Stochastic Environments with High Intrinsic Uncertainty

For intuitive comparison between optimism and randomized criterion, we design **p-DLTV**, a perturbed variant of DLTV, where coefficient $c_t$ is multiplied by a normal distribution $\mathcal{N}(0, 1^2)$. Every experimental setup, pseudocodes, and implementation details can be found in Appendix C.

**N-Chain with high intrinsic uncertainty.** We extend N-Chain environment [27] with stochastic reward to evaluate action selection methods. A schematic diagram of the stochastic N-Chain environment is depicted in Figure 3. The reward is only given in the leftmost and rightmost states and the game terminates when one of the reward states is reached. We set the leftmost reward as $\mathcal{N}(10, 0.1^2)$ and the rightmost reward as $\frac{1}{2}\mathcal{N}(5, 0.1^2) + \frac{1}{2}\mathcal{N}(13, 0.1^2)$ which has a lower mean as 9 but higher variance. The agent always starts from the middle state $s_2$ and should move toward the leftmost state $s_0$ to achieve the greatest expected return. For each state, the agent can take one of six available actions: left, right, and 4 no-op actions. The optimal policy with respect to mean is to move left twice from the start. We set the discount factor $\gamma = 0.9$ and the coefficient $c = 50$.

Despite the simple configuration, the possibility to obtain higher reward in suboptimal state than the optimal state makes it difficult for an agent to determine which policy is optimal until it experiences enough to discern the characteristics of each distribution. Thus, the goal of our toy experiment is to evaluate how rapidly each algorithm could find a risk-neutral optimal policy. The results of varying the size of variance are reported in Appendix D.1.

**Analysis of Experimental Results.** As we design the mean of each return is intended to be similar, examining the learning behavior of the empirical return distribution for each algorithm can provide fruitful insights. Figure 4 shows the empirical PDF of return distribution by using Gaussian kernel density estimation. In Figure 4(b), DLTV fails to estimate the true optimal return distribution. While the return of $(s_2, \texttt{right})$ (red line) is correctly estimated toward the ground truth, $(s_2, \texttt{left})$ (blue line) does not capture the shape and mean due to the lack of experience. At 20K

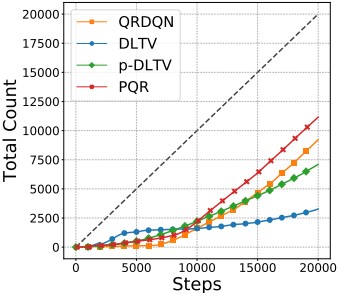

Figure 5: Total count of performing true optimal action. The oracle (dashed line) is to perform the optimal action from start to end.

timestep, the agent begins to see other actions, but the monotonic scheduling already makes the decision like exploitation. Hence, decaying schedule of optimism is not a way to solve the underlying problem. Notably, p-DLTV made a much better estimate than DLTV only by changing from optimism to a randomized scheme. In comparison, PQR estimates the ground truth much better than other baselines with much closer mean and standard-deviation.

Figure 5 shows the number of timesteps when the optimal policy was actually performed to see the interference of biased criterion. Since the optimal policy consists of the same index $a_1$, we plot the total count of performing the optimal action with 10 seeds. From the slope of each line, it is observed that DLTV selects the suboptimal action even if the optimal policy was initially performed. In contrast, p-DLTV avoids getting stuck by randomizing criterion and eventually finds the true optimal policy. The experimental results demonstrate that randomizing the criterion is a simple but effective way for exploration on training process.

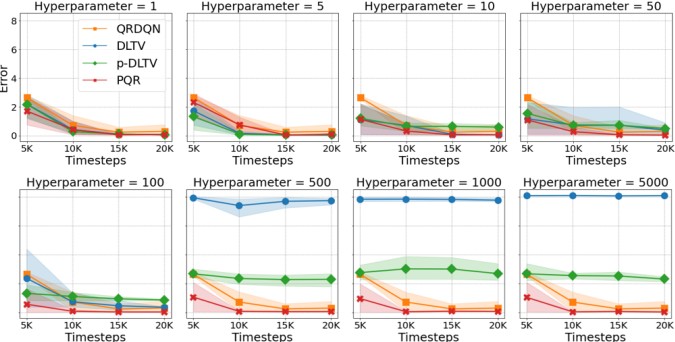

Figure 6: 2-Wasserstein distance between the empirical return distribution and the ground truth $\mathcal{N}(8.1, 0.081^2)$. We use QR-DQN with a fixed setting of $\epsilon$-greedy as a reference baseline, because the hyperparameter of $\epsilon$-greedy is not related to the scale of Q-values.

**Hyperparameter Sensitivity.** In Figure 6, we compute the 2-Wasserstein distance from the ground truth return distribution $\mathcal{N}(10\gamma^2, (0.1\gamma^2)^2)$. Except for QR-DQN , each initial hyperparameter $\{c, \Delta_0\}$ was implemented with grid search on $[1, 5, 10, 50, 100, 500, 1000, 5000]$ in 5 different seeds. As the hyperparameter decreases, each agent is likely to behave as exploitation. One interesting aspect is that, while it may be difficult for DLTV and p-DLTV to balance the scale between the return and bonus term, PQR shows robust performance to the initial hyperparameter. This is because the distorted return is bounded by the support of return distribution, so that PQR implicitly tunes the scale of exploration. In practice, we set $\Delta_0$ to be sufficiently large. See Table2 in Appendix C.1.

## 4.2 Full Atari Results

We compare our algorithm to various DistRL baselines, which have demonstrated good performance on RL benchmarks. In Table 1, we evaluated 55 Atari results, averaging over 5 different seeds at 50M frames. We compared with the published score of QR-DQN [13], IQN [12], and Rainbow [16] via the report of DQN-Zoo [29] and Dopamine [4] benchmark for reliability. This comparison is particularly noteworthy since our proposed method only applys perturbation-based exploration strategy and outperforms advanced variants of QR-DQN. [2]

Table 1: Mean and median of best scores across 55 Atari games, measured as percentages of human baseline. Reference values are from Quan and Ostrovski [29] and Castro et al. [4].

| 50M Performance | Mean | Median | > human | > DQN |
|---|---|---|---|---|
| **DQN-zoo (no-ops)** | 314% | 55% | 18 | 0 |
| **DQN-dopamine (sticky)** | 401% | 51% | 15 | 0 |
| **QR-DQN-zoo (no-ops)** | 559% | 118% | 29 | 47 |
| **QR-DQN-dopamine (sticky)** | 562% | 93% | 27 | 46 |
| **IQN-zoo (no-ops)** | 902% | 131% | 21 | 50 |
| **IQN-dopamine (sticky)** | 940% | 124% | 32 | 51 |
| **RAINBOW-zoo (no-ops)** | 1160% | 154% | 37 | 52 |
| **RAINBOW-dopamine (sticky)** | 965% | 123% | 35 | 53 |
| **PQR-zoo (no-ops)** | 1121% | 124% | 33 | 53 |
| **PQR-dopamine (sticky)** | 962% | 123% | 35 | 51 |

---

[2]In Dopamine framework, IQN was implemented with $n-$step updates with $n = 3$, which improves performance.

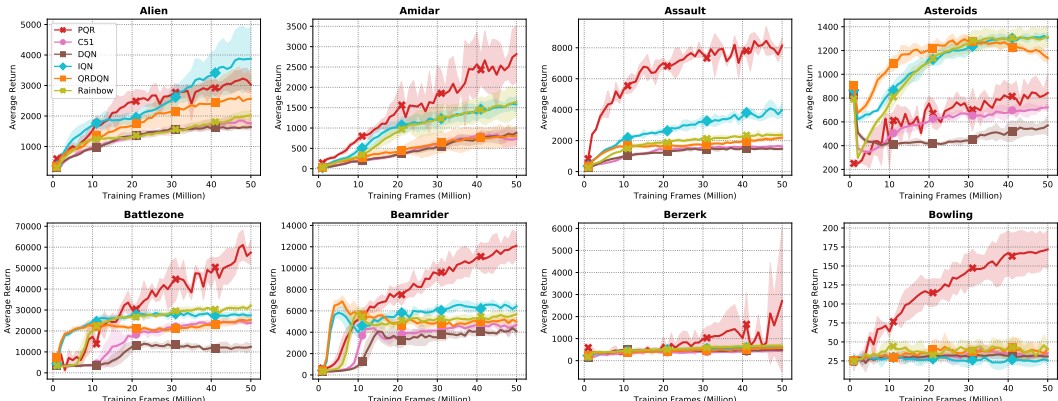

Figure 7: Evaluation curves on 8 Atari games with 3 random seeds for 50 million frames following *sticky actions* protocol [20]. Reference values are from Castro et al. [4].

**No-ops Protocol.** First, we follow the evaluation protocol of [1, 22] on full set of Atari games implemented in OpenAI's Gym [3]. Even if it is well known that the *no-ops* protocol does not provide enough stochasticity to avoid memorization, intrinsic uncertainty still exists due to the *random frame skipping* [20]. While PQR cannot enjoy the environmental stochasticity by the deterministic dynamics of Atari games, PQR achieved 562% performance gain in the mean of human-normalized score over QR-DQN, which is comparable results to Rainbow. From the raw scores of 55 games, PQR wins 39 games against QR-DQN and 34 games against IQN.

**Sticky actions protocol.** To prevent the deterministic dynamics of Atari games, Machado et al. [20] proposes injecting stochasticity scheme, called *sticky actions*, by forcing to repeat the previous action with probability $p = 0.25$. Sticky actions protocol prevents agents from relying on memorization and allows robust evaluation. In Figure 7, PQR shows steeper learning curves, even without any support of advanced schemes, such as $n$-step updates for Rainbow or IQN. In particular, PQR dramatically improves over IQN and Rainbow in ASSAULT, BATTLEZONE, BEAMRIDER, BERZERK and BOWLING. In Table 1, PQR shows robust median score against the injected stochasticity.

It should be noted that IQN benefits from the generalized form of distributional outputs, which reduces the approximation error from the number of quantiles output. Compare to IQN, PQR does not rely on prior distortion risk measure such as CVaR [7], Wang [37] or CPW [36], but instead randomly samples the risk measure and evaluates it with a risk-neutral criterion. Another notable difference is that PQR shows the better or competitive performance solely through its **exploration strategies**, compared to $\epsilon$-greedy baselines, such as QR-DQN, IQN, and especially Rainbow. Note that Rainbow enjoys a combination of several orthogonal improvements such as double Q-learning, prioritized replay, dueling networks, and $n$-step updates.

## 5 Related Works

Randomized or perturbation-based exploration has been focused due to its strong empirical performance and simplicity. In tabular RL, Osband et al. [28] proposed randomized least-squares value iteration (RLSVI) using random perturbations for statistically and computationally efficient exploration. Ishfaq et al. [17] leveraged the idea into optimistic reward sampling by perturbing rewards and regularizers. However, existing perturbation-based methods requires tuning of the hyperparameter for the variance of injected Gaussian noise and depend on well-crafted feature vectors in advance. On the other hand, PDBOO does not rely on the scale of rewards or uncertainties due to the built-in scaling mechanism of risk measures. Additionally, we successfully extend PQR to deep RL scenarios in distributional lens, where feature vectors are not provided, but learned during training.

## 6 Conclusions

In this paper, we proposed a general framework of perturbation in distributional RL which is based on the characteristics of a return distribution. Without resorting to a pre-defined risk criterion,

we revealed and resolved the underlying problem where one-sided tendency on risk can lead to biased action selection under the stochastic environment. To our best knowledge, this paper is the first attempt to integrate risk-sensitivity and exploration by using time-varying Bellman objective with theoretical analysis. In order to validate the effectiveness of PQR, we evaluate on various environments including 55 Atari games with several distributional RL baselines. Without separating the two uncertainties, the results show that perturbing the risk criterion is an effective approach to resolve the biased exploration. We believe that PQR can be combined with other distributional RL or risk-sensitive algorithms as a perturbation-based exploration method without sacrificing their original objectives.

## 7 Acknowledgements

This work was supported in part by National Research Foundation of Korea (NRF, 2021R1A2C2014504(20%) and 2021M3F3A2A02037893(20%)), in part by Institute of Information & communications Technology Planning & Evaluation (IITP) grant funded by the Ministry of Science and ICT (MSIT) (2021-0-00106(15%), 2021-0-02068(15%), 2021-0-01080(15%), and 2021-0-01341(15%)), and in part by AI Graduate School Program, CAU, INMAC and BK21 FOUR program.

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

# A Proof

## A.1 Technical Lemma

Before proving our theoretical results, we present two inequalities for supremum to clear the description.

1. $\sup\limits_{x \in X} |f(x) + g(x)| \leq \sup\limits_{x \in X} |f(x)| + \sup\limits_{x \in X} |g(x)|$

2. $\left| \sup\limits_{x \in X} f(x) - \sup\limits_{x' \in X} g(x') \right| \leq \sup\limits_{x, x' \in X} |f(x) - g(x')|$

*Proof of 1.* Since $|f(x) + g(x)| \leq |f(x)| + |g(x)|$ holds for all $x \in X$,

$$\sup\limits_{x \in X} |f(x) + g(x)| \leq \sup\limits_{x \in X} (|f(x)| + |g(x)|)$$
$$\leq \sup\limits_{x \in X} |f(x)| + \sup\limits_{x \in X} |g(x)|$$

∎

*Proof of 2.* Since $\left| \|a\| - \|b\| \right| \leq \|a - b\|$ for any norm $\| \cdot \|$ and for a large enough $M$,

$$\sup\limits_{x, x' \in X} |f(x) - g(x')| \geq \sup\limits_{x \in X} |f(x) - g(x)|$$
$$= \sup\limits_{x \in X} |(f(x) + M) - (g(x) + M)|$$
$$\geq \left| \sup\limits_{x \in X} (f(x) + M) - \sup\limits_{x \in X} (g(x) + M) \right|$$
$$= \left| \sup\limits_{x \in X} f(x) - \sup\limits_{x' \in X} g(x') \right|$$

∎

## A.2 Proof of Theorem A.3

**Theorem A.3.** If $\xi_t$ converges to 1 in probability on $\Omega$, then $\mathbb{E}\mathcal{T}_{\xi_t}$ converges to $\mathbb{E}\mathcal{T}$ uniformly on $\mathcal{Z}$ for all $s \in \mathcal{S}$ and $a \in \mathcal{A}$.

*Proof.* Recall that $\mathcal{Z} = \left\{ Z : \mathcal{S} \times \mathcal{A} \to \mathscr{P}(\mathbb{R}) \big| \, \mathbb{E}[|Z(s,a)|] \leq V_{\max}, \forall (s,a) \right\}$. Then for any $Z \in \mathcal{Z}$ and $\xi \in \Xi$,

$$\mathbb{E}[|\mathcal{T}_\xi Z|] \leq R_{\max} + \gamma \frac{R_{\max}}{1 - \gamma} = \frac{R_{\max}}{1 - \gamma} = V_{\max}.$$

which implies PDBOO is closed in $\mathcal{Z}$, i.e. $\mathcal{T}_\xi Z \in \mathcal{Z}$ for all $\xi \in \Xi$. Hence, for any sequence $\xi_t$, $Z^{(n)} = \mathcal{T}_{\xi_{n:1}} Z \in \mathcal{Z}$ for any $n \geq 0$.

Since $\xi_t$ converges to 1 in probability on $\Omega$, there exists $T$ such that for any $\epsilon, \delta > 0$ and $t > T$,

$$\mathbb{P}(\Omega_t) := \mathbb{P}\left( \left\{ w \in \Omega : \sup\limits_{w \in \Omega} |\xi_t(w) - 1| \geq \epsilon \right\} \right) \leq \delta$$

For any $Z \in \mathcal{Z}$, $s \in \mathcal{S}$, $a \in \mathcal{A}$, and $t > T$, by using Hölder's inequality,

$$\sup\limits_{Z \in \mathcal{Z}} \sup\limits_{s,a} |\mathbb{E}_{\xi_t}[Z(s,a)] - \mathbb{E}[Z(s,a)]| = \sup\limits_{Z \in \mathcal{Z}} \sup\limits_{s,a} \left| \int_{w \in \Omega} (1 - \xi_t(w)) Z(s,a,w) \mathbb{P}(dw) \right|$$
$$= \sup\limits_{Z \in \mathcal{Z}} \sup\limits_{s,a} \left| \int_{w \in \Omega_t} (1 - \xi_t(w)) Z(s,a,w) \mathbb{P}(dw) + \int_{w \in \Omega \backslash \Omega_t} (1 - \xi_t(w)) Z(s,a,w) \mathbb{P}(dw) \right|$$
$$\leq \mathbb{P}(\Omega_t) \sup\limits_{w \in \Omega_t} |\xi_t(w) - 1| \, V_{\max} + \mathbb{P}(\Omega \backslash \Omega_t) \sup\limits_{w \in \Omega \backslash \Omega_t} |\xi_t(w) - 1| \, V_{\max}$$
$$\leq \delta |B_\xi - 1| V_{\max} + \epsilon V_{\max}$$

which implies that $\mathbb{E}_{\xi_t}$ converges to $\mathbb{E}$ uniformly on $\mathcal{Z}$ for all $s, a$.

By using A.1, we can get the desired result.

$$\sup_{Z \in \mathcal{Z}} \sup_{s,a} |\mathbb{E}[\mathcal{T}_{\xi_t} Z(s,a)] - \mathbb{E}[\mathcal{T} Z(s,a)]|$$

$$\leq \sup_{Z \in \mathcal{Z}} \sup_{s,a} |\mathbb{E}[\mathcal{T}_{\xi_t} Z(s,a)] - \mathbb{E}_{\xi_t}[\mathcal{T}_{\xi_t} Z(s,a)]| + \sup_{Z \in \mathcal{Z}} \sup_{s,a} |\mathbb{E}_{\xi_t}[\mathcal{T}_{\xi_t} Z(s,a)] - \mathbb{E}[\mathcal{T} Z(s,a)]|$$

$$\leq (\delta|B_\xi - 1|V_{\max} + \epsilon V_{\max}) + \gamma \sup_{Z \in \mathcal{Z}} \sup_{s,a} \mathbb{E}_{s'} \left[ \left| \sup_{a'} \mathbb{E}_{\xi_t}[Z(s',a')] - \sup_{a''} \mathbb{E}[Z(s',a'')] \right| \right]$$

$$\leq (\delta|B_\xi - 1|V_{\max} + \epsilon V_{\max}) + \gamma \sup_{Z \in \mathcal{Z}} \sup_{s',a'} |\mathbb{E}_{\xi_t}[Z(s',a')] - \mathbb{E}[Z(s',a')]|$$

$$\leq (\delta|B_\xi - 1|V_{\max} + \epsilon V_{\max}) + \gamma(\delta|B_\xi - 1|V_{\max} + \epsilon V_{\max})$$

$$= (1 + \gamma)(\delta|B_\xi - 1|V_{\max} + \epsilon V_{\max}).$$

∎

## A.3 Proof of Theorem 3.3

**Theorem 3.3.** Let $\xi_n$ be sampled from $\bar{\mathcal{U}}_{\Delta_n}(Z^{(n-1)})$ for every iteration. If Assumption 3.2 holds, then the expectation of any composition of operators $\mathbb{E}\mathcal{T}_{\xi_{n:1}}$ converges, i.e. $\mathbb{E}\mathcal{T}_{\xi_{n:1}}[Z] \to \mathbb{E}[Z^*]$

Moreover, the following bound holds,

$$\sup_{s,a} \left| \mathbb{E}[Z^{(n)}(s,a)] - \mathbb{E}[Z^*(s,a)] \right| \leq \sum_{k=n}^{\infty} \left( 2\gamma^{k-1} V_{\max} + 2 \sum_{i=1}^{k} \gamma^i (\Delta_{k+2-i} + \Delta_{k+1-i}) \right).$$

*Proof.* We denote $a_i^*(\xi_n) = \underset{a'}{\mathrm{argmax}}\, \mathbb{E}_{\xi_n}[Z_i^{(n-1)}(s',a')]$ as the greedy action of $Z_i^{(n-1)}$ under perturbation $\xi_n$. Also, we denote $\underset{s,a}{\sup}|\cdot|$ which is the supremum norm over $s$ and $a$ as $\|\cdot\|_{sa}$.

Before we start from the term $\left\| \mathbb{E}[Z^{(k+1)}] - \mathbb{E}[Z^{(k)}] \right\|_{sa}$, for a given $(s,a)$,

$$\left| \mathbb{E}[Z^{(k+1)}(s,a)] - \mathbb{E}[Z^{(k)}(s,a)] \right|$$

$$\leq \gamma \sup_{s'} \left| \mathbb{E}[Z^{(k)}(s',a^*(\xi_{k+1}))] - \mathbb{E}[Z^{(k-1)}(s',a^*(\xi_k))] \right|$$

$$\leq \gamma \sup_{s'} \left( \left| \mathbb{E}[Z^{(k)}(s',a^*(\xi_{k+1}))] - \max_{a'} \mathbb{E}[Z^{(k)}(s',a')] \right| + \left| \max_{a'} \mathbb{E}[Z^{(k)}(s',a')] - \max_{a'} \mathbb{E}[Z^{(k-1)}(s',a')] \right| \right.$$

$$\left. + \left| \max_{a'} \mathbb{E}[Z^{(k-1)}(s',a')] - \mathbb{E}[Z^{(k-1)}(s',a^*(\xi_k))] \right| \right)$$

$$\leq \gamma \sup_{s',a'} \left| \mathbb{E}[Z^{(k)}(s',a')] - \mathbb{E}[Z^{(k-1)}(s',a')] \right| + \gamma \sum_{i=k-1}^{k} \sup_{s'} \left| \mathbb{E}[Z^{(i)}(s',a^*(\xi_{i+1}))] - \max_{a'} \mathbb{E}[Z^{(i)}(s',a')] \right|$$

$$\leq \gamma \left\| \mathbb{E}[Z^{(k)}] - \mathbb{E}[Z^{(k-1)}] \right\|_{sa} + \gamma \sum_{i=k-1}^{k} \left[ \sup_{s'} \left( \left| \mathbb{E}[Z^{(i)}(s',a^*(\xi_{i+1}))] - \mathbb{E}_{\xi_{i+1}}[Z^{(i)}(s',a^*(\xi_{i+1}))] \right| \right. \right.$$

$$\left. \left. + \left| \max_{a'} \mathbb{E}_{\xi_{i+1}}[Z^{(i)}(s',a')] - \max_{a''} \mathbb{E}[Z^{(i)}(s',a'')] \right| \right) \right]$$

$$\leq \gamma \left\| \mathbb{E}[Z^{(k)}] - \mathbb{E}[Z^{(k-1)}] \right\|_{sa} + 2\gamma \sum_{i=k-1}^{k} \sup_{s',a'} \left( \left| \mathbb{E}[Z^{(i)}(s',a')] - \mathbb{E}_{\xi_{i+1}}[Z^{(i)}(s',a')] \right| \right)$$

$$\leq \gamma \left\| \mathbb{E}[Z^{(k)}] - \mathbb{E}[Z^{(k-1)}] \right\|_{sa} + 2\gamma \sum_{i=k-1}^{k} \Delta_{i+1}$$

where we use A.1.1 in third and fifth line and A.1.2 in sixth line.

Taking a supremum over $s$ and $a$, then for all $k > 0$,

$$\left\|\mathbb{E}[Z^{(k+1)}] - \mathbb{E}[Z^{(k)}]\right\|_{sa} \leq \gamma \left\|\mathbb{E}[Z^{(k)}] - \mathbb{E}[Z^{(k-1)}]\right\|_{sa} + 2 \sum_{i=k-1}^{k} \gamma \Delta_{i+1}$$

$$\leq \gamma^2 \left\|\mathbb{E}[Z^{(k-1)}] - \mathbb{E}[Z^{(k-2)}]\right\|_{sa} + 2 \sum_{i=k-2}^{k-1} \gamma^2 \Delta_{i+1} + 2 \sum_{i=k-1}^{k} \gamma \Delta_{i+1}$$

$$\vdots$$

$$\leq \gamma^k \left\|\mathbb{E}[Z^{(1)}] - \mathbb{E}[Z]\right\|_{sa} + 2 \sum_{i=1}^{k} \gamma^i (\Delta_{k+2-i} + \Delta_{k+1-i})$$

$$\leq 2\gamma^k V_{\max} + 2 \sum_{i=1}^{k} \gamma^i (\Delta_{k+2-i} + \Delta_{k+1-i})$$

Since $\sum_{i=1}^{\infty} \gamma^i = \frac{\gamma}{1-\gamma} < \infty$ and $\sum_{i=1}^{\infty} \Delta_i < \infty$ by assumption, we have

$$\sum_{i=1}^{k} \gamma^i \Delta_{k+1-i} \to 0$$

which is resulted from the convergence of Cauchy product of two sequences $\{\gamma^i\}$ and $\{\Delta_i\}$. Hence, $\{\mathbb{E}[Z^{(k)}]\}$ is a Cauchy sequence and therefore converges for every $Z \in \mathcal{Z}$.

Let $\mathbb{E}[Z^*]$ be the limit point of the sequence $\{\mathbb{E}[Z^{(n)}]\}$. Then,

$$\left\|\mathbb{E}[Z^*] - \mathbb{E}[Z^{(n)}]\right\|_{sa} = \lim_{l \to \infty} \left\|\mathbb{E}[Z^{(n+l)}] - \mathbb{E}[Z^{(n)}]\right\|_{sa}$$

$$\leq \sum_{k=n}^{\infty} \left\|\mathbb{E}[Z^{(k+1)}] - \mathbb{E}[Z^{(k)}]\right\|_{sa}$$

$$= \sum_{k=n}^{\infty} \left(2\gamma^k V_{\max} + 2 \sum_{i=1}^{k} \gamma^i (\Delta_{k+2-i} + \Delta_{k+1-i})\right).$$

$\blacksquare$

### A.4 Proof of Theorem 3.4

**Theorem 3.4.** If $\{\Delta_n\}$ follows the assumption in Theorem 3.3, then $\mathbb{E}[Z^*]$ is the unique solution of Bellman optimality equation.

*Proof.* The proof follows by linearity of expectation. Denote the Q-value based operator as $\bar{\mathcal{T}}$. Note that $\Delta_n$ converges to 0 with regularity of $\mathcal{Z}$ implies that $\xi_n$ converges to 1 in probability on $\Omega$, i.e.,

$$\lim_{n \to \infty} \sup_{s,a} \left| \int_{w \in \Omega} Z^{(n)}(w; s, a)(1 - \xi_n(w)) \mathbb{P}(dw) \right| = 0$$

$$\implies \lim_{n \to \infty} \mathbb{P}\left(\{w \in \Omega : |1 - \xi_n(w)| \geq \epsilon\}\right) = 0$$

By Theorem A.3, for a given $\epsilon > 0$, there exists a constant $K = \max(K_1, K_2)$ such that for every $k \geq K_1$,

$$\sup_{Z \in \mathcal{Z}} \|\bar{\mathcal{T}}_{\xi_k} \mathbb{E}[Z] - \bar{\mathcal{T}} \mathbb{E}[Z]\|_{sa} \leq \frac{\epsilon}{2}.$$

Since $\bar{\mathcal{T}}$ is continuous, for every $k \geq K_2$,

$$\|\bar{\mathcal{T}} \mathbb{E}[Z^{(k)}] - \bar{\mathcal{T}} \mathbb{E}[Z^*]\|_{sa} \leq \frac{\epsilon}{2}.$$

Thus, it holds that

$$\|\bar{\mathcal{T}}_{\xi_{k+1}}\mathbb{E}[Z^{(k)}] - \bar{\mathcal{T}}\mathbb{E}[Z^*]\|_{sa} \leq \|\bar{\mathcal{T}}_{\xi_{k+1}}\mathbb{E}[Z^{(k)}] - \bar{\mathcal{T}}\mathbb{E}[Z^{(k)}]\|_{sa} + \|\bar{\mathcal{T}}\mathbb{E}[Z^{(k)}] - \bar{\mathcal{T}}\mathbb{E}[Z^*]\|_{sa}$$

$$\leq \sup_{Z \in \mathcal{Z}} \|\bar{\mathcal{T}}_{\xi_{k+1}}\mathbb{E}[Z] - \bar{\mathcal{T}}\mathbb{E}[Z]\|_{sa} + \|\bar{\mathcal{T}}\mathbb{E}[Z^{(k)}] - \bar{\mathcal{T}}\mathbb{E}[Z^*]\|_{sa}$$

$$\leq \frac{\epsilon}{2} + \frac{\epsilon}{2}$$

$$= \epsilon.$$

Therefore, we have

$$\mathbb{E}[Z^*] = \lim_{k \to \infty} \mathbb{E}[Z^{(k)}] = \lim_{k \to \infty} \mathbb{E}[Z^{(k+1)}] = \lim_{k \to \infty} \mathbb{E}[\mathcal{T}_{\xi_{k+1}} Z^{(k)}] = \lim_{k \to \infty} \bar{\mathcal{T}}_{\xi_{k+1}}\mathbb{E}[Z^{(k)}] = \bar{\mathcal{T}}\mathbb{E}[Z^*]$$

Since the standard Bellman optimality operator has a unique solution, we derived the desired result. ∎

## B   Algorithm Pipeline

Figure 8 shows the pipeline of our algorithm. With the schedule of perturbation bound $\{\Delta_n\}$, the ambiguity set $\mathcal{U}_{\Delta_n}(Z_{n-1})$ can be defined by previous $Z_{n-1}$. For each step, (distributional) perturbation $\xi_n$ is sampled from $\mathcal{U}_{\Delta_n}(Z_{n-1})$ by the symmetric Dirichlet distribution and then PDBOO $\mathcal{T}_{\xi_n}$ can be performed.

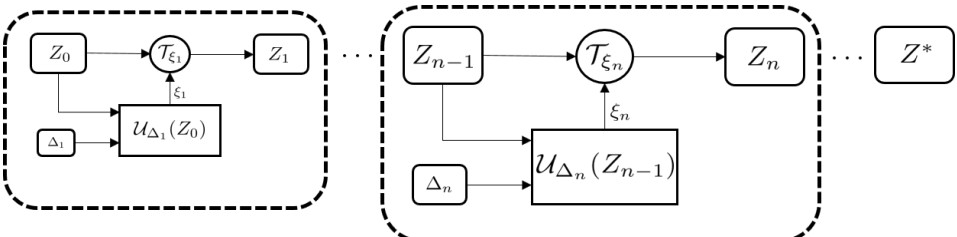

Figure 8: Pipeline of PDBOO.

# C   Implementation details

Except for each own hyperparameter, our algorithms and DLTV shares the same hyperparameter and network architecture with QR-DQN [13] for a fair comparison. Also, we set up p-DLTV by only multiplying a gaussian noise $\mathcal{N}(0,1)$ to the coefficient of DLTV. We **do not combine** any additional improvements of Rainbow such as double Q-learning, dueling network, prioritized replay, and $n$-step update. Experiments on LunarLander-v2 and Atari games were performed with 3 random seeds. The training process is 0-2% slower than QR-DQN due to the sampling $\xi$ and reweighting procedures.

## C.1   Hyperparameter Setting

We report the hyperparameters for each environments we used in our experiments.

Table 2: Table of hyperparameter setting

| Hyperparameters | N-Chain | LunarLander | Atari Games |
|---|---|---|---|
| Batch size | 64 | 128 | 32 |
| Number of quantiles | 200 | 170 | 200 |
| $n-$step updates | | 1 | |
| Network optimizer | | Adam | |
| $\beta$ | | Grid search[0.05, 0.1, 0.5, 1] $\times \mathbf{1}^N$ | |
| $\kappa$ | | 1 | |
| Memory size | 1e6 | 1e5 | 1e6 |
| Learning rate | 5e-5 | 1.5e-3 | 5e-5 |
| $\gamma$ | 0.9 | 0.99 | 0.99 |
| Update interval | 1 | 1 | 4 |
| Target update interval | 25 | 1 | 1e4 |
| Start steps | 5e2 | 1e4 | 5e4 |
| $\epsilon$ (train) | | LinearAnnealer($1 \rightarrow$ 1e-2) | |
| $\epsilon$ (test) | 1e-3 | 1e-3 | 1e-3 |
| $\epsilon$ decay steps | 2.5e3 | 1e5 | 2.5e5 |
| Coefficient $c$ | | Grid search[1e0, 5e0, 1e1, 5e1, 1e2, 5e2, 1e3, 5e3] | |
| $\Delta_0$ | 5e2 | 5e4 | 1e6 |
| Number of seeds | 10 | 3 | 3 |

## C.2   Pseuodocode of p-DLTV

---
**Algorithm 2** Perturbed DLTV (p-DLTV)

---
  **Input:** transition $(s, a, r, s')$, discount $\gamma \in [0, 1)$
  $Q(s', a') = \frac{1}{N} \sum_j \theta_j(s', a')$
  $c_t \sim c\,\mathcal{N}(0, \frac{\ln t}{t})$                                              // Randomize the coefficient
  $a^* \leftarrow \mathrm{argmax}_{a'}(Q(s', a') + c_t \sqrt{\sigma_+^2(s', a')})$
  $\mathcal{T}\theta_j \leftarrow r + \gamma\theta_j(s', a^*), \quad \forall j$
  **Output:** $\sum_{i=1}^N \mathbb{E}_j[\rho_{\hat{\tau}_i}^\kappa(\mathcal{T}\theta_j - \theta_i(s, a))]$

---

# D   Further experimental results & Discussion

## D.1   N-Chain

To explore the effect of intrinsic uncertainty, we run multiple experiments with various reward settings for the rightmost state as keeping their mean at 9. As the distance between two Gaussians was increased, the performance of DLTV decrease gradually, while other algorithms show consistent results. The result implies the interference of one-sided tendency on risk is proportional to the magnitude of the intrinsic uncertainty and the randomized criterion is effective in escaping from the issue.

Table 3: Total counts of performing true optimal action with 4 different seeds.

| Total Count | (8,10) | (7,11) | (6,12) | (5,13) | (4,14) | (3,15) | (2,16) | (1,17) |
|---|---|---|---|---|---|---|---|---|
| QR-DQN | 12293 | 11381 | 11827 | 12108 | 10041 | 11419 | 9696 | 11619 |
| DLTV | 9997 | 9172 | 9646 | 9251 | 7941 | 6964 | 7896 | 7257 |
| p-DLTV | 14344 | 14497 | 13769 | **15507** | 14469 | **14034** | 14068 | 13404 |
| PQR | **14546** | **15018** | **14693** | 15142 | **15361** | 13859 | **14602** | **14354** |

## D.2 LunarLander-v2

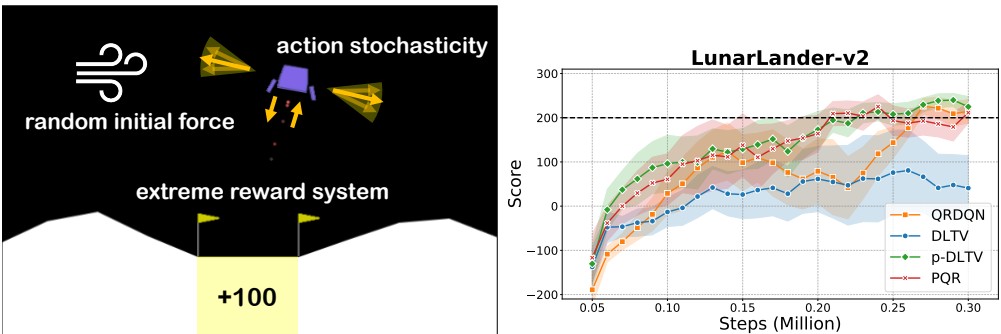

Figure 9: (**Left**) Three main environmental factors causing high intrinsic uncertainty on LunarLander-v2. (**Right**) Performance on LunarLander-v2

To verify the effectiveness of the proposed algorithm in the complex environment with **high intrinsic uncertainty**, we conduct the experiment on LunarLander-v2. We have focused on three main factors that increase the intrinsic uncertainty from the structural design of LunarLander environment:

- **Random initial force:** The lander starts at the top center with an random initial force.
- **Action stochasticity:** The noise of engines causes different transitions with same action.
- **Extreme reward system:** If the lander crashes, it receives -100 points. If the lander comes to rest, it receives +100 points.

Therefore, several returns with a fixed policy have a high variance. As previously discussed about the fixedness from N-Chain environment, we can demonstrate that randomized approaches, PQR and p-DLTV, outperform other baselines in LunarLander-v2.

## D.3 Atari games

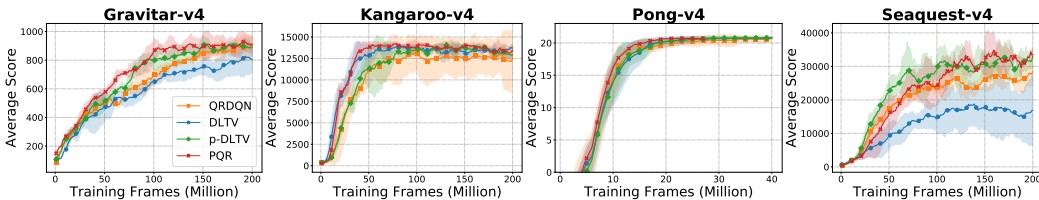

Figure 10: Evaluation curves on Atari games. All curves are smoothed over 10 consecutive steps with three random seeds. In case of Pong-v4, we resize the x-axis, since it can easily obtain the optimal policy with few interactions due to its environmental simplicity.

We test our algorithm under 30 no-op settings to align with previous works. We compare our baseline results with results from the DQN Zoo framework [29], which provides the full benchmark results on 55 Atari games at 50M and 200M frames. We report the average of the best scores over 5 seeds for each baseline algorithms up to 50M frames.

However, recent studies tried to follow the setting proposed by Machado et al. [20] for reproducibility, where they recommended using sticky actions. Hence, we provide all human normalized scores results across 55 Atari games for 50M frames including previous report of Dopamine and DQN Zoo framework to help the follow-up researchers as a reference. We exclude `Defender` and `Surround` which is not reported on Yang et al. [38] because of relialbility issues in the Dopamine framework. In summary,

- DQN Zoo framework corresponds to 30 no-op settings (version **v4**).

- Dopamine framework corresponds to sticky actions protocol (version **v0**).

### D.3.1   Ablation on PQR schedules

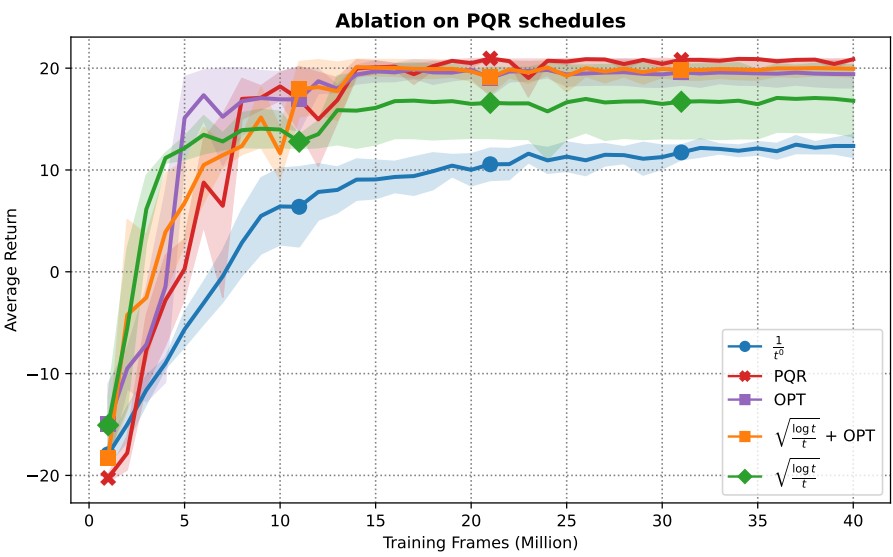

Figure 11: Evaluation curves on Pong-v4 environments.

To investigate the effect of the schedule of $\Delta_t$, we run the experiment on Pong-v4 and set up several baselines as follows:

- $1/t^0$ : A fixed size ambiguity set. $\Delta_t = O(1)$

- PQR : Our main algorithm. $\Delta_t = O(1/t^{1+\epsilon})$

- OPT : We fix the output vector, sampled from the Dirichlet distribution, as $[0, 0, \dots, 1]$, forcing the agent to estimate only optimistically.

- $\sqrt{\log t/t}$ : We imitate the schedule of p-DLTV (which does not satisfy the sufficient condition we presented). $\Delta_t = O(\sqrt{\log t/t})$

- $\sqrt{\log t/t}$ + OPT : We imitate the schedule of DLTV (which does not satisfy the sufficient condition we presented). We fixed the output vector, sampled from the Dirichlet distribution, as $[0, 0, \dots, 1]$, forcing the agent to estimate only optimistically. $\Delta_t = |O(\sqrt{\log t/t})|$

In this experiment, our proposed PQR is the only method that stably achieves the maximum score with low variance. In the case of optimism (purple, orange curve), the agent learns quickly in the early stages, but converges without reaching the maximum score. In the case of fixed ambiguity set (blue curve), it converges to suboptimal and eventually shows low performance. This result implies the necessity of time-varying schedule of $\Delta_t$. Finally, when imitating the schedule of p-DLTV (green curve), the performance also degrades implying that the proposed sufficient condition is quite tight.

## D.4 Comparison with QUOTA

Zhang and Yao [40] have proposed Quantile Option Architecture(QUOTA) which derives different policies corresponding to different risk levels and consider them as options. By using an option-based framework, the agent learns a high-level policy that adaptively selects a pessimistic or optimistic exploration strategy. While QUOTA has a similar approach in high-level idea, PQR gives a lot of improvements in both theoretical analysis and experimental results.

- **Theoretical guarantees of convergence toward risk-neutrality.**

  Since the agent selects via randomized risk criterion, the natural question is "How should we control the injected randomness without sacrificing the original purpose of risk-neutrality?". In this work, we provide the sufficient condition for convergence without sacrificing risk-neutral perspective. Although QUOTA explores by using optimism or pessimism of a value distribution, there is no discussion whether the convergence is guaranteed toward a risk-neutral objective.

- **Explaining the effectiveness of randomized strategy.**

  QUOTA tested on two Markov chains to illustrate the inefficiency of expectation-based RL. It assumed that each task has an inherent, but unknown, preferred risk strategy, so agents should learn hidden preference. In contrast, we point out that the amount of inherent (intrinsic) uncertainty causes the inefficiency of fixed optimism or pessimism based exploration.

- **Significant performance difference in experimental results.**

  QUOTA is based on option-based learning which requires an additional option-value network. While QUOTA aims to control risk-sensitivity by transforming into an option $O$, the introduction of an option-value network requires the agent to explore an action space $|O| \times |A|$. This opposes the idea of efficient exploration as a factor that increases the complexity of learning. In contrast, PQR does not require a additional network and explores over the original action space. In addition, PQR does not artificially discretize the ambiguity set of risk measurement. Another main reason is that PQR does not depend on an greedy schedule which is well-known for inefficient exploration strategies in tabular episodic MDP [18]. PQR solely explores its own strategies which is a simple yet effective approach. However, QUOTA depends on a greedy schedule in both quantile and option networks.

## D.5 Reproducibility issues on DLTV

For the expected concerns about the comparison with DLTV, we address some technical issues to correct misconceptions of their performance. Before we reproduce the empirical results of DLTV, Mavrin et al. [21] did not report each raw scores of Atari games, but only the relative performance with cumulative rewards comparing with QR-DQN. While DLTV was reported to have a cumulative reward 4.8 times greater than QR-DQN, such gain mainly comes from VENTURE which is evaluated as 22,700% from their metric (i.e., 463% performance gain solely). However, the approximate raw score of VENTURE was 900 which is lower than our score of 993.3. Hence, the report with cumulative rewards causes a severe misconception that can be overestimated where the human-normalized score is commonly used for evaluation metrics. For a fair comparison, we computed based on mean and median of human-normalized scores and obtained results of 603.66% and 109.90%. Due to the absence of public results, however, DLTV was inevitably excluded from the comparison with human-normalized score in the main paper for reliability. In Table 4 and 7, we report our raw scores and human-normalized score of DLTV based on QR-DQN_zoo performance.

Table 4: Performance comparison on QUOTA, DLTV, and PQR.

| QUOTA >QR-DQN_Zhang | QR-DQN_zoo >QR-DQN_Zhang | PQR >QUOTA | PQR >QR-DQN_Zhang | PQR >DLTV |
|---|---|---|---|---|
| 30 | 34 | 42 | 42 | 39 |
| **Avg HN Score(QR-DQN_zoo)** | **Avg HN Score(QR-DQN_Zhang)** | **Avg HN Score(QUOTA)** | **Avg HN Score(DLTV)** | **Avg HN Score(PQR)** |
| 505.02 | 463.47 | 383.70 | 603.66 | 1078.00 |
| **Med HN Score(QR-DQN_zoo)** | **Med HN Score(QR-DQN_Zhang)** | **Med HN Score(QUOTA)** | **Med HN Score(DLTV)** | **Med HN Score(PQR)** |
| 120.74 | 78.07 | 91.08 | 109.90 | 129.25 |

Table 5: Raw scores across all 55 games, starting with 30 no-op actions. We report the best scores for DQN, QR-DQN, IQN and Rainbow on 50M frames, averaged by 5 seeds. Reference values were provided by DQN Zoo framework [29]. **Bold** are wins against DQN, QR-DQN and IQN, and *asterisk are wins over Rainbow.

| GAMES | RANDOM | HUMAN | DQN(50M) | QR-DQN(50M) | IQN(50M) | RAINBOW(50M) | PQR(50M) |
|---|---|---|---|---|---|---|---|
| Alien | 227.8 | 7127.7 | 1541.5 | 1645.7 | 1769.2 | 4356.9 | **2455.8** |
| Amidar | 5.8 | 1719.5 | 324.2 | 683.4 | 799.2 | 2549.2 | **938.4** |
| Assault | 222.4 | 742.0 | 2387.8 | 11684.2 | 15152.4 | 9737.0 | 10759.2 |
| Asterix | 210.0 | 8503.3 | 5249.5 | 18373.4 | 32598.2 | 33378.6 | 10490.5 |
| Asteroids | 719.1 | 47388.7 | 1106.3 | 1503.9 | 1972.6 | 1825.4 | 1662.0 |
| Atlantis | 12850.0 | 29028.1 | 283392.2 | 937275.0 | 865360.0 | 941740.0 | 897640.0 |
| BankHeist | 14.2 | 753.1 | 389.0 | 1223.9 | 1266.8 | 1081.7 | 1038.8 |
| BattleZone | 2360.0 | 37187.5 | 19092.4 | 26325.0 | 30253.9 | 35467.1 | 28470.5 |
| BeamRider | 363.9 | 16926.5 | 7133.1 | 12912.0 | 19251.4 | 15421.9 | 10224.9 |
| Berzerk | 123.7 | 2630.4 | 577.4 | 826.5 | 918.9 | 2061.6 | *137873.1 |
| Bowling | 23.1 | 160.7 | 34.4 | 45.4 | 41.5 | 54.7 | *86.9 |
| Boxing | 0.1 | 12.1 | 87.2 | 99.6 | 99.2 | 99.8 | 97.1 |
| Breakout | 1.7 | 30.5 | 316.8 | 426.5 | 468.0 | 335.3 | 380.3 |
| Centipede | 2090.9 | 12017.0 | 4935.7 | 7124.0 | 7008.3 | 5691.4 | *7291.2 |
| ChopperCommand | 811.0 | 7387.8 | 974.2 | 1187.8 | 1549.0 | 5525.1 | 1300.0 |
| CrazyClimber | 10780.5 | 35829.4 | 96939.0 | 93499.1 | 127156.5 | 160757.7 | 84390.9 |
| DemonAttack | 152.1 | 1971.0 | 8325.6 | 106401.8 | 110773.1 | 85776.5 | 73794.0 |
| DoubleDunk | -18.6 | -16.4 | -15.7 | -10.5 | -12.1 | -0.3 | **-7.5** |
| Enduro | 0.0 | 860.5 | 750.6 | 2105.7 | 2280.6 | 2318.3 | *2341.2 |
| FishingDerby | -91.7 | -38.7 | 8.2 | 25.7 | 23.4 | 35.5 | **31.7** |
| Freeway | 0.0 | 29.6 | 24.4 | 33.3 | 33.7 | 34.0 | **34.0** |
| Frostbite | 65.2 | 4334.7 | 408.2 | 3859.2 | 5650.8 | 9672.6 | 4148.2 |
| Gopher | 257.6 | 2412.5 | 3439.4 | 6561.9 | 26768.9 | 32081.3 | *47054.5 |
| Gravitar | 173.0 | 3351.4 | 180.5 | 548.1 | 470.2 | 2236.8 | **635.8** |
| Hero | 1027.0 | 30826.4 | 9948.3 | 9909.8 | 12491.1 | 38017.9 | **12579.2** |
| IceHockey | -11.2 | 0.9 | -11.4 | -2.1 | -4.2 | 1.9 | **-1.4** |
| Jamesbond | 29.0 | 302.8 | 486.4 | 1163.8 | 1058.0 | 14415.5 | **2121.8** |
| Kangaroo | 52.0 | 3035.0 | 6720.7 | 14558.2 | 14256.0 | 14383.6 | *14617.1 |
| Krull | 1598.0 | 2665.5 | 7130.5 | 9612.5 | 9616.7 | 8328.5 | *9746.1 |
| KungFuMaster | 258.5 | 22736.3 | 21330.9 | 27764.3 | 39450.1 | 30506.9 | *43258.6 |
| MontezumaRevenge | 0.0 | 4753.3 | 0.3 | 0.0 | 0.2 | 80.0 | 0.0 |
| MsPacman | 307.3 | 6951.6 | 2362.9 | 2877.5 | 2737.4 | 3703.4 | **2928.9** |
| NameThisGame | 2292.3 | 8049.0 | 8196.4 | 11843.3 | 11582.2 | 11341.5 | 10298.2 |
| Phoenix | 761.4 | 7242.6 | 10153.6 | 35128.6 | 29138.9 | 49138.8 | 20453.8 |
| Pitfall | -229.4 | 6463.7 | -9.5 | 0.0 | 0.0 | 0.0 | **0.0** |
| Pong | -20.7 | 14.6 | 18.7 | 20.9 | 20.9 | 21.0 | **21.0** |
| PrivateEye | 24.9 | 69571.3 | 266.6 | 100.0 | 100.0 | 160.0 | *372.4 |
| Qbert | 163.9 | 13455.0 | 5567.9 | 12808.4 | 15101.8 | 24484.9 | 15267.4 |
| Riverraid | 1338.5 | 17118.0 | 6782.8 | 9721.9 | 13555.9 | 17522.9 | 11175.3 |
| RoadRunner | 11.5 | 7845.0 | 29137.5 | 54276.3 | 53850.9 | 52222.6 | 50854.7 |
| Robotank | 2.2 | 11.9 | 31.4 | 54.5 | 53.8 | 64.5 | **60.3** |
| Seaquest | 68.4 | 42054.7 | 2525.8 | 7608.2 | 17085.6 | 3048.9 | *19652.5 |
| Skiing | -17098.1 | -4336.9 | -13930.8 | -14589.7 | -19191.1 | -15232.3 | *-9299.3 |
| Solaris | 1236.3 | 12326.7 | 2031.5 | 1857.3 | 1301.5 | 2522.6 | *2640.0 |
| SpaceInvaders | 148.0 | 1668.7 | 1179.1 | 1753.2 | 2906.7 | 2715.3 | 1749.4 |
| StarGunner | 664.0 | 10250.0 | 24532.5 | 63717.3 | 78503.4 | 107177.8 | 62920.6 |
| Tennis | -23.8 | -8.3 | -0.9 | 0.0 | 0.0 | 0.0 | -1.0 |
| TimePilot | 3568.0 | 5229.2 | 2091.8 | 6266.8 | 6379.1 | 12082.1 | **6506.4** |
| Tutankham | 11.4 | 167.6 | 138.7 | 210.2 | 204.4 | 194.3 | *231.3 |
| UpNDown | 533.4 | 11693.2 | 6724.5 | 27311.3 | 35797.6 | 65174.2 | 36008.1 |
| Venture | 0.0 | 1187.5 | 53.3 | 12.5 | 17.4 | 1.1 | *993.3 |
| VideoPinball | 16256.9 | 17667.9 | 140528.4 | 104405.8 | 341767.5 | 465636.5 | **465578.3** |
| WizardOfWor | 563.5 | 4756.5 | 3459.9 | 14370.2 | 10612.1 | 12056.1 | 6132.8 |
| YarsRevenge | 3092.9 | 54576.9 | 16433.7 | 21641.4 | 21645.0 | 67893.3 | **27674.4** |
| Zaxxon | 32.5 | 9173.3 | 3244.9 | 9172.1 | 8205.2 | 22045.8 | **10806.6** |

Table 6: Raw scores across 55 games. We report the best scores for DQN, QR-DQN, IQN*, and Rainbow on 50M frames, averaged by 5 seeds. Reference values were provided by Dopamine framework [4]. **Bolds** are wins against DQN, QR-DQN, and *asterisk are wins over IQN* and Rainbow. **Note that IQN* and Rainbow implemented in Dopamine framework applied $n$-step updates with $n = 3$ which improves performance.**

| GAMES | RANDOM | HUMAN | DQN(50M) | QR-DQN(50M) | IQN*(50M) | RAINBOW(50M) | PQR(50M) |
|---|---|---|---|---|---|---|---|
| Alien | 227.8 | 7127.7 | 1688.1 | 2754.2 | 4016.3 | 2076.2 | **3173.9** |
| Amidar | 5.8 | 1719.5 | 888.2 | 841.6 | 1642.8 | 1669.6 | ***2814.7** |
| Assault | 222.4 | 742.0 | 1615.9 | 2233.1 | 4305.6 | 2535.9 | ***8456.5** |
| Asterix | 210.0 | 8503.3 | 3326.1 | 3540.1 | 7038.4 | 5862.3 | ***19004.6** |
| Asteroids | 719.1 | 47388.7 | 828.2 | 1333.4 | 1336.3 | 1345.1 | 851.8 |
| Atlantis | 12850.0 | 29028.1 | 388466.7 | 879022.0 | 897558.0 | 870896.0 | **880303.7** |
| BankHeist | 14.2 | 753.1 | 720.2 | 964.1 | 1082.8 | 1104.9 | **1050.1** |
| BattleZone | 2360.0 | 37187.5 | 15110.3 | 25845.6 | 29959.7 | 32862.1 | ***61494.4** |
| BeamRider | 343.9 | 16926.5 | 4771.3 | 7143.0 | 7113.7 | 6331.9 | ***12217.6** |
| Berzerk | 123.7 | 2630.4 | 529.2 | 603.2 | 627.3 | 697.8 | ***2707.2** |
| Bowling | 23.1 | 160.7 | 38.5 | 55.3 | 33.6 | 55.0 | ***174.1** |
| Boxing | 0.1 | 12.1 | 80.0 | 96.6 | 97.8 | 96.3 | **96.7** |
| Breakout | 1.7 | 30.5 | 113.5 | 40.7 | 164.4 | 69.8 | 48.5 |
| Centipede | 2090.9 | 12017.0 | 3403.7 | 3562.5 | 3746.1 | 5087.6 | ***31079.8** |
| ChopperCommand | 811.0 | 7387.8 | 1615.9 | 1600.3 | 6654.1 | 5982.0 | **4653.9** |
| CrazyClimber | 10780.5 | 35829.4 | 111493.8 | 108493.9 | 131645.8 | 135786.1 | 105526.0 |
| DemonAttack | 152.1 | 1971.0 | 4396.7 | 3182.6 | 7715.5 | 6346.4 | ***19530.2** |
| DoubleDunk | -18.6 | -16.4 | -16.7 | 7.4 | 20.2 | 17.4 | **15.0** |
| Enduro | 0.0 | 860.5 | 2268.1 | 2062.6 | 766.5 | 2255.6 | 1765.5 |
| FishingDerby | -91.7 | -38.7 | 12.3 | 48.4 | 41.9 | 37.6 | 46.8 |
| Freeway | 0.0 | 29.6 | 25.8 | 33.5 | 33.5 | 33.2 | 33.0 |
| Frostbite | 65.2 | 4334.7 | 760.2 | 8022.8 | 7824.9 | 5697.2 | ***8401.5** |
| Gopher | 257.6 | 2412.5 | 3495.8 | 3917.1 | 11192.6 | 7102.1 | ***12252.9** |
| Gravitar | 173.0 | 3351.4 | 250.7 | 821.3 | 1083.5 | 926.2 | 703.5 |
| Hero | 1027.0 | 30826.4 | 12316.4 | 14980.0 | 18754.0 | 31254.8 | **15655.8** |
| IceHockey | -11.2 | 0.9 | -6.7 | -4.5 | 0.0 | 2.3 | **0.0** |
| Jamesbond | 29.0 | 302.8 | 500.0 | 802.3 | 1118.8 | 656.7 | ***1454.9** |
| Kangaroo | 52.0 | 3035.0 | 6768.2 | 4727.3 | 11385.4 | 13133.1 | ***13894.0** |
| Krull | 1598 | 2665.5 | 6181.1 | 8073.9 | 8661.7 | 6292.5 | ***31927.4** |
| KungFuMaster | 258.5 | 22736.3 | 20418.8 | 20988.3 | 33099.9 | 26707.0 | **22040.4** |
| MontezumaRevenge | 0.0 | 4753.3 | 2.6 | 300.5 | 0.7 | 501.2 | 0.0 |
| MsPacman | 307.3 | 6951.6 | 2727.2 | 3313.9 | 4714.4 | 3406.4 | ***5426.5** |
| NameThisGame | 2292.3 | 8049.0 | 5697.3 | 7307.9 | 9432.8 | 9389.5 | ***9891.3** |
| Phoenix | 761.4 | 7245.6 | 5833.7 | 4641.1 | 5147.2 | 8272.9 | 5260 |
| Pitfall | -229.4 | 6463.7 | -16.8 | -3.4 | -0.4 | 0.0 | ***0.0** |
| Pong | -20.7 | 14.6 | 13.2 | 19.2 | 19.9 | 19.4 | **19.7** |
| PrivateEye | 24.9 | 69571.3 | 1884.6 | 680.7 | 1287.3 | 4298.8 | ***12806.1** |
| Qbert | 163.9 | 13455.0 | 8216.2 | 17228.0 | 15045.5 | 17121.4 | 15806.9 |
| Riverraid | 1338.5 | 17118.0 | 9077.8 | 13389.4 | 14868.6 | 15748.9 | **14101.3** |
| RoadRunner | 11.5 | 7845.0 | 39703.1 | 44619.2 | 50534.1 | 51442.4 | **48339.7** |
| Robotank | 2.2 | 11.9 | 25.8 | 53.6 | 65.9 | 63.6 | 48.7 |
| Seaquest | 68.4 | 42054.7 | 1585.9 | 4667.9 | 20081.3 | 3916.2 | **5038.1** |
| Skiing | -17098.1 | -4336.9 | -17038.2 | -14401.6 | -13755.6 | -17960.1 | ***-9021.2** |
| Solaris | 1236.3 | 12326.7 | 2029.5 | 2361.7 | 2234.5 | 2922.2 | ***7145.3** |
| SpaceInvaders | 148.0 | 1668.7 | 1361.1 | 940.2 | 3115.0 | 1908.0 | **1602.4** |
| StarGunner | 664.0 | 10250.0 | 1676.5 | 23593.3 | 60090.0 | 39456.3 | **59404.6** |
| Tennis | -23.8 | -9.3 | -0.1 | 19.2 | 3.5 | 0.0 | ***15.4** |
| TimePilot | 3568.0 | 5229.2 | 3200.9 | 6622.8 | 9820.6 | 9324.4 | 5597.0 |
| Tutankham | 11.4 | 167.6 | 138.8 | 209.9 | 250.4 | 252.2 | 147.3 |
| UpNDown | 533.4 | 11693.2 | 10405.6 | 29890.1 | 44327.6 | 18790.7 | **32155.5** |
| Venture | 0.0 | 1187.5 | 50.8 | 1099.6 | 1134.5 | 1488.9 | 1000.0 |
| VideoPinball | 16256.9 | 17667.9 | 216042.7 | 250650.0 | 486111.5 | 536364.4 | **460860.9** |
| WizardOfWor | 563.5 | 4756.5 | 2664.9 | 2841.8 | 6791.4 | 7562.7 | **5738.2** |
| YarsRevenge | 3092.9 | 54576.9 | 20375.7 | 66055.9 | 57960.3 | 31864.4 | ***67545.8** |
| Zaxxon | 32.5 | 9173.3 | 1928.6 | 8177.2 | 12048.6 | 14117.5 | **9531.8** |

Table 7: Raw scores across all 49 games, starting with 30 no-op actions. We report the best scores for QR-DQN_zoo[29], QR-DQN_Zhang[40](implemented by QUOTA to evaluate the relative improvement) for a fair comparison and QUOTA[40], DLTV[21] on 40M frames, averaged by 3 seeds. **Bold** are wins against QUOTA and DLTV.

| Games | Random | Human | QR-DQN_zoo(40M) | QR-DQN_Zhang(40M) | QUOTA(40M) | DLTV(40M) | PQR(40M) |
|---|---|---|---|---|---|---|---|
| Alien | 227.8 | 7127.7 | 1645.7 | 1760.0 | 1821.9 | 2280.9 | **2406.9** |
| Amidar | 5.8 | 1719.5 | 552.9 | 567.9 | 571.4 | 1042.7 | 644.1 |
| Assault | 222.4 | 742 | 9880.4 | 3308.7 | 3511.1 | 5896.2 | **10759.2** |
| Asterix | 210 | 8503.3 | 13157.2 | 6176.0 | 6112.1 | 6336.6 | **8431.0** |
| Asteroids | 719.1 | 47388.7 | 1503.9 | 1305.3 | 1497.6 | 1268.7 | 1416.00 |
| Atlantis | 12850 | 29028.1 | 750190.1 | 978385.3 | 965193.0 | 845324.9 | 897640.0 |
| BankHeist | 14.2 | 753.1 | 1146.1 | 644.7 | 735.2 | 1183.7 | 1038.8 |
| BattleZone | 2360 | 37187.5 | 17788.4 | 22725.0 | 25321.6 | 23315.8 | **28470.5** |
| BeamRider | 363.9 | 16926.5 | 10684.2 | 5007.8 | 5522.6 | 6490.1 | **10224.9** |
| Bowling | 23.1 | 160.7 | 44.3 | 27.6 | 34.0 | 29.8 | **86.9** |
| Boxing | 0.1 | 12.1 | 98.2 | 95.0 | 96.1 | 112.8 | 97.1 |
| Breakout | 1.7 | 30.5 | 401.5 | 322.1 | 316.7 | 260.9 | **357.7** |
| Centipede | 2090.9 | 12017.0 | 6633.0 | 4330.3 | 3537.9 | 4676.7 | **6803.6** |
| ChopperCommand | 811.0 | 7387.8 | 1133.1 | 3421.1 | 3793.0 | 2586.3 | 1500.0 |
| CrazyClimber | 10780.5 | 35829.4 | 93499.1 | 107371.6 | 113051.7 | 92769.1 | 83900.0 |
| DemonAttack | 152.1 | 1971.0 | 98026.6 | 80026.6 | 61005.1 | 146928.9 | 73794.0 |
| DoubleDunk | -18.6 | -16.4 | -10.5 | -21.6 | -21.5 | -23.3 | **-10.5** |
| Enduro | 0.0 | 860.5 | 2105.7 | 1220.0 | 1162.3 | 5665.9 | 2252.8 |
| FishingDerby | -91.7 | -38.7 | 25.7 | -9.6 | -59.0 | -8.2 | **31.7** |
| Freeway | 0.0 | 29.6 | 30.9 | 30.6 | 31.0 | 34.0 | **34.0** |
| Frostbite | 65.2 | 4334.7 | 3822.7 | 2046.3 | 2208.5 | 3867.6 | **4051.2** |
| Gopher | 257.6 | 2412.5 | 4191.2 | 9443.8 | 6824.3 | 10199.4 | **47054.5** |
| Gravitar | 173.0 | 3351.4 | 477.4 | 414.3 | 457.6 | 357.9 | **583.6** |
| IceHockey | -11.2 | 0.9 | -2.4 | -9.8 | -9.9 | -14.3 | **-2.1** |
| Jamesbond | 29.0 | 302.8 | 907.1 | 601.7 | 495.5 | 779.8 | **1747.1** |
| Kangaroo | 52.0 | 3035 | 14171 | 2364.6 | 2555.8 | 4596.7 | **14385.1** |
| Krull | 1598.0 | 2665.5 | 9618.2 | 7725.4 | 7747.5 | 10012.21 | 9537.0 |
| KungFuMaster | 258.5 | 22736.3 | 27576.5 | 17807.4 | 20992.5 | 23078.4 | **38074.1** |
| MontezumaRevenge | 0.0 | 4753.3 | 0.0 | 0.0 | 0.0 | 0.0 | 0.0 |
| MsPacman | 307.3 | 6951.6 | 2561.0 | 2273.3 | 2423.5 | 3191.7 | 2895.6 |
| NameThisGame | 2292.3 | 8049.0 | 11770.0 | 7748.2 | 7327.5 | 8368.1 | **10298.2** |
| Pitfall | -229.4 | 6463.7 | 0.0 | -32.9 | -30.7 | - | **0.0** |
| Pong | -20.7 | 14.6 | 20.9 | 19.6 | 20.0 | 21.0 | **21.0** |
| PrivateEye | 24.9 | 69571.3 | 100.0 | 419.3 | 114.1 | 1358.6 | **372.4** |
| Qbert | 163.9 | 13455.0 | 8348.2 | 10875.3 | 11790.2 | 15856.2 | 14593.0 |
| Riverraid | 1338.5 | 17118.0 | 8814.1 | 9710.4 | 10169.8 | 10487.3 | 9374.7 |
| RoadRunner | 11.5 | 7845.0 | 52575.7 | 27640.7 | 27872.2 | 49255.7 | 44341.0 |
| Robotank | 2.2 | 11.9 | 50.4 | 45.1 | 37.6 | 58.4 | 53.9 |
| Seaquest | 68.4 | 42054.7 | 5854.6 | 1690.5 | 2628.6 | 3103.8 | **16011.2** |
| SpaceInvaders | 148.0 | 1668.7 | 1281.8 | 1387.6 | 1553.8 | 1498.6 | **1562.6** |
| StarGunner | 664.0 | 10250.0 | 53624.7 | 49286.6 | 52920.0 | 53229.5 | **55475.0** |
| Tennis | -23.8 | -8.3 | 0.0 | -22.7 | -23.7 | -18.4 | **-1.0** |
| TimePilot | 3568.0 | 5229.2 | 6243.4 | 6417.7 | 5125.1 | 6931.1 | 6506.4 |
| Tutankham | 11.4 | 167.6 | 200.0 | 173.2 | 195.4 | 130.9 | **213.3** |
| UpNDown | 533.4 | 11693.2 | 22248.8 | 30443.6 | 24912.7 | 44386.7 | 33786.3 |
| Venture | 0.0 | 1187.5 | 12.5 | 5.3 | 26.5 | 1305.0 | 0.0 |
| VideoPinball | 16256.9 | 17667.9 | 104227.2 | 123425.4 | 44919.1 | 93309.6 | **443870.0** |
| WizardOfWor | 563.5 | 4756.5 | 13133.8 | 5219.0 | 4582.0 | 9582.0 | 6132.8 |
| Zaxxon | 32.5 | 9173.3 | 7222.7 | 6855.1 | 8252.8 | 6293.0 | **10250.0** |

