# OpenReview forum: "Pitfall of Optimism: Distributional Reinforcement Learning by Randomizing Risk Criterion"
_NeurIPS.cc/2023/Conference — NeurIPS 2023 poster_

### Official Review · Reviewer_rSdX · 2023-07-02

**Soundness:** 3 good
**Presentation:** 3 good
**Contribution:** 3 good
**Rating:** 6
**Confidence:** 4

**Summary:**

This paper proposes a distributional RL algorithm, PQR, that uses distributional return estimates for exploration by updating with a greedy distributional Bellman operator for time-varying, random risk measures. The paper demonstrates that, under a simple concentration condition on the random risk measures over time, the means of return distributions learned by PQR converge to the optimal Q-values. It is argued that the use of random risk measures (as opposed to static risk measures) produces updates that are less biased away from the optimal Q-values. It is shown that PQR demonstrates favorable exploration performance in a simple chain MDP and in Atari.

**Strengths:**

The paper presents a well-motivated algorithm for leveraging return distributions for exploration in (deep) RL. The idea of using random risk measures in greedy distributional Bellman updates to veer less from the optimal Q-values is novel and interesting. Furthermore, the empirical results of the PQR algorithm appear to be quite good.

**Weaknesses:**

The proposed algorithm itself may not be substantially novel -- I suspect similar results would hold for QR-DQN with a dynamic distortion risk measure that tends to the identity sufficiently quickly.

The theoretical results are fairly weak. Firstly, convergence of the return distributions is only established with respect to their first moments. Moreover, the upper bound equation in Theorem 3.3 is difficult to interpret, and I am not sure what one should conclude from this upper bound.

Additionally, the influence of the schedule $(\Delta_t)_{t\geq 0}$ and the Dirichlet parameter $\beta$ is not really studied -- I suspect, at least in theory, that there are more optimal data-based schedules.

## Grammatical / non-major issues

L4 "present a novel distributional reinforcement learning that selects..." -> "present a novel distributional reinforcement learning **algorithm** that selects..."

Last sentence in the abstract cites [6] as a risk-sensitive distributional RL algorithm, but this is not really a distributional RL algorithm.

L144 mentions the perturbed probability distribution $\xi\mathbb{P}$, but this doesn't really type-check. $\mathbbf{P}$ is a probability measure (maps measurable sets to probabilities) and $\xi$ is a probability density function (mapping elements of the sample space to reals), so the product $\xi\mathbb{P}$ is not exactly correct.

Theorem 3.4 needs to be made more precise. It makes a claim about the fixed point of the Bellman optimality operator, but I think you meant the PDBOO.

**Questions:**

L172 says "PDBOO has a significant distinction in that it performs dynamic programming that adheres to the risk-neutral optimal policy..." -- what does it mean to "adhere to the risk-neutral optimal policy"? The PDBOO updates are still performing greedy updates w.r.t. risk-sensitive policies.

As mentioned above, convergence of return distributions is not really established at all under the PDBOO update -- we only know about the convergence of their means. But if the dynamics of the other statistics of the return distributions are essentially unknown, how can you be sure that the random risk measures are actually estimating anything meaningful?

Is there any correspondence between the PDBOO targets and the "Thompson Sampling" approach introduced by Riou and Honda (2020) http://proceedings.mlr.press/v117/riou20a/riou20a.pdf? There may be an interesting connection there, which could perhaps help add precision to the statement about "adhering to the risk-neutral optimal policy".

**Limitations:**

Limitations are addressed adequately.

---

> ### Author Rebuttal · Authors · 2023-08-09
>
> We would like to thank you for the time and effort in reviewing our paper. Please find below our response to the main points raised in the review.
>
> ## The proposed algorithm itself may not be substantially novel.
> Of course, any scheduling that is sufficiently close to the QR-DQN can yield a risk-neutral optimal solution, but this is not sufficient for exploration. Since our theory provides sufficient conditions for satisfying the risk-neutrality, we have made it as exploratory as possible by choosing scheduling that satisfies non-trivial boundary conditions. We also believe that our proof of convergence for the time-varying bellman operator is sufficiently novel.
>
> ## Convergence of the return distributions is only established with respect to their first moments.
> I think this is a very important question. In general, the return distribution in distRL does not have a unique fixed distribution. This is because there can be multiple greedy policies, since they are defined only for expectations and do not focus on the shape of the distribution. If we could define a total ordering on the set of greedy policies,(e.g., select the action with the lowest variance if expected values are equal) then there is a unique fixed distribution [1,2].
>
> The same is true for the situation we face, and the total ordering is likewise required for convergence at the distribution level. If the total ordering is well-defined so that the uniqueness of the fixed distribution is guaranteed, then our fixed distribution is the same as the solution of the standard distributional Bellman optimality equation. We will add more details in the revised version in Section 3.2.
>
> [1] Bellemare, Marc G., Will Dabney, and Rémi Munos. "A distributional perspective on reinforcement learning." International conference on machine learning. PMLR, 2017.
>
> [2] Bellemare, Marc G., Will Dabney, and Mark Rowland. Distributional reinforcement learning. MIT Press, 2023.
>
> ## The influence of the schedule $\Delta_t$ and $\beta$
> We have analyzed the effect of learning as a function of the size of the initial value of $\Delta_t$ and PQR shows consistent results unlike other algorithms. **In the global response, we additionally experiment with different forms of scheduling for PQR.**
>
> For the hyperparameter $\beta$, we chose a weight of the form $c * \textbf{1}^N$ to draw symmetrically from the simplex. If the value of $c$ is too large, the sampled weights will be around the center of the simplex, which is not much different from the original mean operator. So we have experimented with the setting $c=grid search[0.05, 0.1, 0.5, 1]$. We then chose a small value so that the perturbation gap is large enough by sampling $\xi$ in the form of spikes at the edge of the simplex.
>
> ## Reference [6] is not really a distributional RL algorithm
> Thank you for the careful review. We'll fix the citation in that paragraph and include another reference.
>
> ## Expression issue of $\xi\mathcal{P}$
> This was a sloppy expression. We'll fix it in the main text as follows,
> - line 144: which can be interpreted as the expectation of $X$ under perturbed probability distribution $Q(dw) = \xi(w)P(dw)$.
>
> ## Fixed point of the Bellman optimality operator in Theorem 3.4 seems to mean the PDBOO.
> Since Theorem 3.4 shows that the fixed point of PDBOO is the same as that of a standard DBOO, this statement reflects our meaning well. We state this once more in lines 193-194 and 197.
>
> ## Connection between the PDBOO targets and the Thompson Sampling?
> We agree that the suggested method is similar to our approach. Thanks for suggesting the reference, it seems to be quite an important paper. The proposed nonparametric TS algorithm seems to have many similarities with our approach in that it uses a weighted sum of Dirichlet distributions for action selection. Their situation is different from ours in that they record all visits and reward histories, but perhaps extending their work to value function approximation approach would deepen the connection. The improved version of nonparametric TS algorithms controls the Dirichlet parameter, while PQR controls the upper bound of the perturbation gap, which seems to have a different approach but similar effect.

---

> > ### Comment · Reviewer_rSdX · 2023-08-14
> >
> > Thanks for the clarifications. I have increased my score.

---

> > > ### Author Response · Authors · 2023-08-15
> > > **Thanks for your positive response**
> > >
> > > We appreciate your positive response, and we're glad that our response was clear. We will incorporate your thoughtful feedback into a revised version.

---

### Official Review · Reviewer_jYGR · 2023-07-03

**Soundness:** 2 fair
**Presentation:** 3 good
**Contribution:** 2 fair
**Rating:** 4
**Confidence:** 4

**Summary:**

The authors propose a new exploration strategy that is more principled than epsilon-greedy or DLTV of Mavrin et al. (based on truncated variance). The proposed exploration strategy is to based on a novel perturbed distributional Bellman optimality operator, where the goal (Eqn. 1) is to minimize the distributional Bellman objective subject to the perturbation being sampled from an uncertainty set; the uncertainty set is defined as importance weighting the next state's return distribution in the Bellman operator, where the radius of the set decreases with time. The authors show that by appropriately decaying the set's radius, the algorithm can learn the optimal value function, and provide experimental validation on N-chain and Atari, showing competitive performance to IQN and Rainbow.

**Strengths:**

1. Perturbing the expectation with a density ratio is a simple but neat idea.
2. Experimental results seem pretty promising, given that QR-DQN's performance significantly improves with the proposed strategy, sometimes even surpassing IQN.

**Weaknesses:**

1. I am not fully sold that perturbing the Bellman operator is a principled way to modeling epistemic uncertainty of our estimates. In particular, the paper seems to suggest that optimism is actually biased and unprincipled, while we know that in both theory and practice, optimism in the face of uncertainty is principled for online RL and minimax optimal in most well-known settings.
1b. I think the paper would benefit from a clearer and more proper explanation of how prior settings are biased / unbiased, and how the proposed approach is better/worse than prior exploration strategies.
2. if I understand correctly, the proposed uncertainty set seems to be (s,a)-independent, which seems wrong. (in particular, if some parts of the state space was not visited often, that uncertainty should be high, while uncertainty should be low for frequented states).

**Questions:**

1. Typically, distributionally robust optimization (DRO) optimizes the worst-case performance. However, in eqn. 1, you sample from the uncertainty set instead of taking a supremum. Can you give some intuition on why you care about the average performance over the set, rather than worst case?
2. Is the perturbation gap related/inspired by the dual form of dynamic risk with coherent risk measures?
3. Theorem 3.3 shows some kind of convergence of the learned $E Z^{(n)}$ to the true $Q*$ , but it seems that as n -> infinity, the bound does not necessarily to go zero. Doesn't this mean there will be some bias in the proposed approach?
4. To be clear, does your implementation only use QR-DQN + the proposed PQR exploration strategy? Do you make use of target networks, double Q-learning, prioritized replay, or any of the bag of tricks that Rainbow uses?
5. In Section 4.2, what is meant by "intrinsic uncertainty" of Atari? Without sticky actions, Atari is fully deterministic and there is no aleatoric uncertainty (only epistemic uncertainty), right?


**Limitations:**

Please see weakness/questions.

---

> ### Author Rebuttal · Authors · 2023-08-09
>
> We would like to thank you for the time and effort in reviewing our paper. Please find below our response to the main points raised in the review.
>
> ## How does the perturbed Bellman operator is a principled way to modeling epistemic uncertainty?
> We are not trying to model epistemic uncertainty through perturbations; rather, we are trying to point out that using the OFU without considering the entanglement of intrinsic uncertainty (commonly referred to as risk) leads to biased exploration and degrades performance.
>
> In principle, OFU works well for most online RL tasks where epistemic uncertainty (e.g., the number of visits) can be modeled. However, as mentioned in lines 41-47, in the deep reinforcement learning case, it is difficult to separate epistemic uncertainty because learning a feature representation of the high-dimensional state-action space and updating the bellman target occur simultaneously. There have been several attempts to solve this problem, and we have proposed a simple yet effective exploration method that does not separate the two uncertainties.
>
> ## Proposed uncertainty set seems to be $(s,a)$-independent, which seems wrong.
> That's a great point. First, we would like to note that our method is not exploration based on parametric uncertainty. The ambiguity set we define is for sampling perturbations xi, which is a weight applied equally to each return distribution, where the perturbation gaps are different for each $(s,a)$ depending on the shape of the distribution. Therefore, we believe that $(s,a)$-dependent elements like bonus correspond to perturbation gaps, but not to perturbation gaps or ambiguity sets.
> Also, note that $\bar{U}_\Delta$ is the definition that the upper bound of the perturbation interval for all $(s,a)$ is less than $\Delta$, which does not mean that all $(s,a)$ have the “same” amount of perturbation gap. Thus, for uncertain state-action which will have high parametric uncertainty, the estimated distribution will also have a high variance, so the perturbation gap will be large and still be chosen frequently.
>
> ## Give some intuition on why we care about the average performance, rather than the worst case.
> If we use a min-max operator with respect to risk, the solution is always more conservative (pessimistic) than the standard solution as it always considers the worst-case scenario. On the other hand, we considered the average-case which has the possibility to maintain risk-neutrality. Specifically, we aimed to find a sufficient condition to ensure risk-neutrality by scheduling the ambiguity set of risks. As such, we only share the definition of ambiguity set by the DRO literature, but with a different goal of risk-neutrality.
>
> ## Relation/Inspiration by the dual form of coherent risk measures?
> Coherent risk measures and perturbation gaps seem to be related, since we try to construct ambiguity sets according to the DRO literature. However, we were not inspired by the dual form and have not yet attempted to make the connection. The definition of the perturbation gap is a natural consequence of obtaining the upper bound of Theorem 3.3. Because it is a tractable and simple form, we were able to design a practical PQR by defining the perturbation gap as shown in the paper. It would be interesting to define the perturbation gap in a more sophisticated way.
>
> ## The bound in Theorem 3.3 does not necessarily go zero.
> We cannot guarantee that delta_t will always go to zero if it is unconstrained. Hence, in Assumption 3.2, we gave sufficient conditions to ensure that the upper bound always goes to zero. Please make sure that the subscript "$k=n$" in the summation of the right terms is not misinterpreted as "$k=1$".
>
> ## Does our work only use the proposed PQR exploration strategy?
> We implemented PQR as you would expect, and we believe it's the right way to make a fair comparison. We did not use any improvement techniques of RAINBOW to verify the effectiveness of our method alone. Please note that our baseline comparison, IQN-dopamine, also used $n=3$ multi-steps and still PQR achieved higher performance.
>
> ## What is meant by “intrinsic uncertainty” of Atari?
> We apologize for not explaining this in more detail. While it's true that the Atari game itself (without no-op or sticky actions) is completely deterministic. However, in v0 and v4, the versions we have experimented with, opengym additionally introduced random frame skipping, so there's another source of intrinsic uncertainty. We will revise the sentence to make this clear.

---

### Official Review · Reviewer_qLSL · 2023-07-12

**Soundness:** 2 fair
**Presentation:** 2 fair
**Contribution:** 2 fair
**Rating:** 5
**Confidence:** 3

**Summary:**

This paper investigates the exploration problem in distributional reinforcement learning (DRL). It proposes the Perturbed Distributional Bellman Operator (PDBOO) as an extension of the distributional Bellman operator, which introduces non-directional noise to the target return distribution. This extension originates from the problem that a one-sided risk tendency can cause biased action selection, which might lead to unsuitable behavior depending on tasks. To address this issue, the PDBOO adds non-directional noise as a perturbation to the return distribution targeted for learning, ensuring risk neutrality. The perturbation term is sampled at each time step. This approach allows for the selection of diverse actions while maintaining risk neutrality. The authors provide a theoretical analysis of PDBOO, demonstrating how the strength of the perturbation should be scheduled to asymptotically converge to the unique fixed point of the Bellman optimality equation. Then, the PDBOO is applied to the Quantile Regression Deep Q Network (QR-DQN) to propose a method called Perturbed Quantile Regression (PQR). The effectiveness of PQR is evaluated in a 4-states chain MDP and 55 Atari games and compared with several DQN variants.

**Strengths:**

- The paper introduces the Perturbed Distributional Bellman Operator (PDBOO), a novel extension of the distributional Bellman operator. This approach enables diverse exploration while maintaining risk neutrality in distributional reinforcement learning. The authors also develop Perturbed Quantile Regression (PQR), a DQN variant that estimates quantiles based on PDBOO. The proposal represents a new and interesting direction of the study in distributional RL.
- The authors provide a solid theoretical analysis of PDBOO, demonstrating how the strength of the perturbation should be scheduled to achieve the unique fixed point of the Bellman optimality equation asymptotically. This result provides confidence in their proposed methods and valuable insights for algorithm development.
- The proposed PQR method is evaluated in a 4-states chain MDP and 55 Atari games. While the results showed room for improvement, the evaluation confirms the potential practical applicability of PQR and its ability to address the exploration problem while maintaining risk neutrality.

**Weaknesses:**

- The paper lacks an explanation as to why the proposed approach is more effective compared to other methods. It is crucial to discuss under what tasks PDBOO is effective and when it might not be suitable to understand its value.
   - For example, PDBOO would add random noise (perturbation) to the target return distribution for learning. Depending on the perspective, this could also have the side effect of making it more challenging to learn the return distribution due to the existence of noise. I would like to know if there are situations where PDBOO should not be applied.
   - Also, it is not clear why the approach of PDBOO is necessary, as it seems that what the authors want to achieve with PDBOO would not be the perturbation of the target return distribution for learning but instead could also be accomplished by perturbing the estimated return distribution at the time of action selection. This is somewhat supported by the results of p-DLTV in the N-chain MDP experiment.

- The overall experimental results seem weak and do not resolve the abovementioned questions. Furthermore, the improved performance of PQR is observed in a subset of games, and the improvement seems modest. There is no comparison with p-DLTV, which showed similar good performance in the N-chain MDP experiment.

**Questions:**

- How exactly does the proposed PQR select actions? Does it use an epsilon-greedy or softmax policy? More importantly, how does it compute the values of the actions which will be the input to the policy?  Does it sample the perturbation \xi, as PDBOO does, and skew the distribution for calculating the mean? I think there are two timings at which the distribution can be perturbed by introducing ξ: during the parameter θ update, and during the action selection. This paper only seems to discuss the former, but considering that p-DLTV performs well similarly to the proposed method, wouldn't it also be valid to perturb the mean for the action selection?
- What is the type of policy for each baseline used in the experiments? Is it epsilon-greedy or softmax policy? I do not see it stated.
- Regarding the N-Chain with high intrinsic uncertainty experiment, the difference in the decaying rate of \Delta and c between PQR and DLTV or p-DLTV might also be significant. What are the authors' thoughts? It would be best to align the experiment conditions as much as possible.
- The authors used the N-chain task where risk-based exploration like DLTV obviously does not work well, so the presented results are reasonable. On the other hand, in a task where DLTV works well (e.g., both s_0 and s_4 have the same reward variance), I think it is essential to experiment to see how the proposed method performs to compare the proposed method fairly.

**Limitations:**

- The paper lacks a thorough discussion of the limitations of the proposed PDBOO and PQR. The authors should provide more insight into scenarios where the method might not be applicable or efficient.
- As pointed out above, why the proposed method is effective (for instance, as opposed to p-DLTV or simply increasing the temperature during action selection) remains unclear. It seems that the questions are not adequately addressed in the experiments. The authors should consider addressing these points to provide a more complete view of their method.

---

> ### Author Rebuttal · Authors · 2023-08-09
>
> Thank you for your feedback and for taking the time to review our submission. Contrary to your concerns, we want to emphasize that we've already designed PDBOO in the direction you're thinking, and we have provided detailed answers below.
>
> ## Misconceptions about how PDBOO is designed
> We will address this concern first, as it is the one that has caused the most misunderstanding. We want to emphasize that, as in p-DLTV, PDBOO only uses the perturbed return to determine the action that maximizes it, and updates the target by the “unperturbed” return distribution of that action. This is mentioned in lines 156-157 and in the pseudocode. To learn in the way you are concerned, the target must be updated by the perturbed pdf of $Z$ as $Q(dw) = \xi(w)P(dw)$. Specifically, our distributional Bellman update is written as
> $T_\xi Z(s,a) = R(s,a) + \gamma Z(S', a^*(\xi))$, not
> $T_\xi Z(s,a) = R(s,a) + \gamma (\xi Z)(S', a^*(\xi))$.
> The latter is probably what you interpreted.
>
> We are deeply concerned that this misunderstanding has had a significant impact on the evaluation of the paper, and we would like to add the following sentence to line 158 to avoid confusion.
>
> - “Specifically, PDBOO perturbs the estimated distribution only to select the optimal behavior, while the target is updated with the original (unperturbed) return distribution.”
>
> **In the global response, we add the comment that the target is updated with unperturbed return distribution.**
>
> ## When is PDBOO effective/not effective?
> PDBOO is more effective in environments with high intrinsic uncertainty, such as stochasticity from transitions, rewards, or observations, and less effective in near-deterministic environments. (However, this is not necessarily a disadvantage.) As an experiment, Appendix D.1 shows the results of varying the amount of intrinsic uncertainty, and we can see that p-DLTV and PQR give consistent results, while DLTV shows a gradual degradation in performance as intrinsic uncertainty increases. This efficiency is crucial to achieve a better Bellman update because the experimental results show that choosing particular sub-optimal actions repeatedly contaminates the return distribution due to its incorrect policy evaluation.
>
> ## Performance improvements in a subset of games and marginal improvements.
> We respectfully disagree with your comment. To prevent performance exploitation from a subset of games, we have already provided 4 different performance metrics including the median of human normalized score following the evaluation protocol of [1] in Table 1. Furthermore, the metrics compared to human and DQN scores support that PQR outperforms IQN and is competitive with RAINBOW. As mentioned in lines 307-309, RAINBOW applied several improvement techniques, while PQR only changed its exploration strategy, which cannot be considered a marginal performance.
>
> [1] Bellemare, Marc G., et al. "The arcade learning environment: An evaluation platform for general agents." Journal of Artificial Intelligence Research 47 (2013): 253-279.
>
> ## The type of policy for each baseline
> Only DLTV, p-DLTV, and PQR explored in their own way without using an epsilon-greedy schedule. DLTV's exploration method is mentioned in lines 102-106. It is noteworthy that PQR, like soft q-learning, includes the exploration process in its Bellman operator. We mentioned in lines 97-99 and 307-308 that the remaining baselines QR-DQN, IQN, and RAINBOW all use epsilon-greedy schedules.
> Align the experiment conditions as much as possible.
> DLTV and p-DLTV have their own schedules, so we don't think it's appropriate to change them. We also set the scheduling of PQR to be close to $\frac{1}{t}$ based on theory. However, we think your concerns are valid, and it would be interesting to experiment with scheduling PQR like DLTV, even if it's not consistent with theory. We'll try to explain the various scheduling aspects in more detail.
>
> ## No experiment where DLTV works well.
> The N-Chain environment we design has been tested in scenarios where the optimal action of the predefined risk measure  is not necessarily the risk-neutral optimal solution (action). Our experiments on the standard benchmarks, 55 Atari games and Lunarlander-v2 have different reward designs, which confirms the generalizability of our action selection method. In other words, to design an environment where DLTV performs better, we need to make the optimal solution of the mean-variance risk measure equal to the risk-neutral optimal solution. This is an artificial design that exploits the property of optimal return distributions that should be unknown.
>
> Despite the above issue of fair comparison, we can answer your comment, “compare the proposed method fairly” in the standard benchmark setting as follows.
> In Figure 10 in Appendix D.3, we demonstrated the comparison on Atari with 30 no-ops protocol. Among the injected stochasticities, it is known that the 30 no-ops has small randomness, because this protocol only has an effect on the beginning of the episode. By the nature of Atari games themselves, Pong has less intrinsic uncertainty because the opponent has the identical policy even if our agent is trained, but Seaquest has different transitions over time as its manual notes
>
> *“so after each round, take a breath - enemy subs and sharks will increase in speed”*.
>
> We now know that Pong has small intrinsic uncertainty, but Seaquest has larger intrinsic uncertainty (different transitions on a given state over time). In Figure 10, DLTV works well in Pong (low intrinsic uncertainty) but shows a lower performance in Seaquest because of the high intrinsic uncertainty.
>
> We want to emphasize that DLTV is a decent algorithm, but our proposed method outperforms in the cases of high intrinsic uncertainty as shown in Figure 7,9,10 and all result tables.

---

> > ### Author Response · Authors · 2023-08-18
> > **Additional Response by Authors**
> >
> > ##  No comparison with p-DLTV
> >
> > We apologize for the delay in writing about this issue due to the character limit. First of all, p-DLTV is an exemplary algorithm for us to compare OFU and randomized approach very effectively. Also, PQR is a more general method compared to p-DLTV which only considers 2nd order moments (variance), whereas PQR uses all the moments indirectly. We experiment with p-DLTV for some Atari environments in Appendix D.3, but do not include it in the main text for three reasons.
> >
> > 1. Reproducibility problem with DLTV: As we wrote in detail in Appendix D, DLTV was difficult to check for reproducibility because it did not provide raw scores. In Table 4, our implementation showed that DLTV had a Human Normalized Mean/Median Score of 603%/109% based on 40M frames, which is a marginal performance difference compared to QR-DQN's 505%/120%. Therefore, we decided to baseline only those algorithms that were reproducible, and DLTV and p-DLTV were excluded from the baseline for a fair comparison.
> >
> > 2. Hyperparameter sensitivity: While experimenting on several Atari games, we found that p-DLTV is very sensitive to its coefficient, c, as shown in Figure 6. As noted in Appendix C.1, we ran a grid search for six values of c, but this suffered from the problem that the optimal value was different for different environments depending on the scale of the reward, making it less effective. PQR, on the other hand, has the advantage of being intrinsically tunable, since xi is defined as a weight that is independent of scale, and thus showed robust performance.
> >
> > 3. Not consistent with our proposed theory: From a theoretical perspective, p-DLTV does not satisfy the sufficient conditions we proposed. Since we want to guarantee risk-neutrality while the agent explores, p-DLTV is only an intermediate algorithm so it is not the final goal. Furthermore, we experimented further in the global response with PQR-$\sqrt{\log t/t }$, which has the same form of scheduling as p-DLTV and found that it easily falls into suboptimality.

---

> > > ### Comment · Reviewer_qLSL · 2023-08-19
> > >
> > > Thank you for the comprehensive explanations and addressing the concerns raised. Having read the comments from the other reviewers and the authors' responses, my understanding of the paper has deepened. I will adjust my score upwards.

---

> > > > ### Author Response · Authors · 2023-08-21
> > > > **Thanks for your positive feedback**
> > > >
> > > > Thank you for your positive response, and we're glad that our response helped you understand better. We'll make sure the revised version is clear in our wording to avoid any misunderstandings.

---

### Official Review · Reviewer_u65W · 2023-07-15

**Soundness:** 3 good
**Presentation:** 3 good
**Contribution:** 3 good
**Rating:** 7
**Confidence:** 4

**Summary:**

In this work, the authors address the issue of biased exploration caused by a one-sided tendency on risk in action selection by proposing the method perturbed quantile regression (PQR). PQR selects actions by randoming risk criterion while retaining a risk-neutral objective. The authors also derive a sufficient condition for the convergence of the proposed Bellman operator without satisfying the conventional contraction property. Results are demonstrated in an extended N-Chain environment and the Atari suite showing improved performance.


**Strengths:**

The paper was written very well in a pedagogical manner which provided intuition and understanding of the challenges in the field surrounding the contribution of the method. Deep technical detail was provided with accessible explanations.

The motivation for the method was well grounded intuitively. The experiment section also directly confirmed the intended contributions, organizing each experiment and result around specific questions directly aligned to the motivation.

The background section was thorough and clear. Since the authors pulled together several topics which many researchers do not study all of, this presentation was very important and added a clear path of accessibility to the technical detail in the rest of the paper.

The supplementary material provided sufficient information to reproduce results including code. The authors also included substantial supporting theoretical information in proofs and a large amount of additional convincing experiments.

The contribution to the community regarding managing risk in exploration strategies is important and interesting.

**Weaknesses:**


The authors claim limitations about epistemic and parametric uncertainty metrics as well as optimism in the face of uncertainty approaches which I thought were not backed up in the paper strongly enough given that these claims are the premise of the entire work. The paragraph in the introduction which presents the limitations describes it as a broad issue, yet only one paper is cited as an example. To provide more context, it would help to give at least more citations here to back up the importance of this challenge. The deeper explanation of the DLTV paper is not necessary for all additional citations.

Please provide more comments in Algorithm 1 (like the current text “Select greedy action…”). The algorithm is fairly clear but with the density of notation, recalling the variable references would be much easier for the reader with textual reminders in the algorithm block.

My primary concerns regarding this work are related to the experiments in Section 4.1 which require clarification. Figure 4 is quite hard to read since the text is very small. Reformatting could make these plots readable when the paper is printed. Furthermore, the results in Figure 4 are very hard to understand due not only to the legibility but also due to insufficient explanation in the experimental results section and figure. See Questions for more information.


**Questions:**

I am unsure if I am correctly interpreting the results in Figure 4. Is the lower standard deviation around the dotted line for the mean a1 considered to be a better estimation result? Given that this experimental setting is new (the authors indicate that they adapt the N-Chain environment and cite work - DQN - which presents very different result metrics), substantially clearer explanation is required about the connection between the presentation and the claims.

**Limitations:**

The paper does not discuss limitations or broader impact. This discussion should be added.

---

> ### Author Rebuttal · Authors · 2023-08-09
>
> We would like to thank you for the time and effort in reviewing our paper. Please find below our response to the main points raised in the review.
>
> ## Provide more context to back up the importance of this challenge.
> We will add a few more papers in line 50 that attempt to use distributions for exploration following DLTV.
>
> - Ramtin Keramati, Christoph Dann, Alex Tamkin, and Emma Brunskill. Being optimistic to be conservative: Quickly learning a cvar policy. In Proceedings of the AAAI Conference on Artificial Intelligence, volume 34, pages 4436–4443, 2020.
>
> ## Provide more comments in Algorithm 1
> Thank you for your consideration. **In the global response, we add more detailed explanations in Algorithm 1**, including comments about scheduling for $\Delta_t$ and refinement to a weighted function.
>
> ## Reformatting and more explanation about Figure 4
> Thanks for the details. We will try to make the font size as large as possible.
> Also, you are right that a smaller standard deviation around $a_1$, the blue curve, means a better estimate. Regarding the explanation of Fig. 4, we will add information about the ground truth return distribution of $s_0$ and $s_4$ to the caption of Fig. 3 to make it easier to check whether the estimate of the red/blue curve is close to the ground truth.
>
> - Ground truth return of $s_0$ : $\gamma^2 \mathcal{N}(10, 0.1^2)$
> - Ground truth return of $s_4$ : $\gamma^2 [\frac{1}{2} \mathcal{N}(5, 0.1^2) + \frac{1}{2} \mathcal{N}(13, 0.1^2)]$

---

> > ### Comment · Reviewer_u65W · 2023-08-14
> > **Reply to Rebuttal**
> >
> > Thank you very much for your additional details and responses to my questions. With these clarifications, I raised my confidence from a 3 to a 4 for my accept rating.

---

> > > ### Author Response · Authors · 2023-08-15
> > > **Thanks for your positive response**
> > >
> > > We appreciate your positive response and we're glad that it has helped to increase your confidence in our paper. We will incorporate your thoughtful feedback into a revised version.

---

### Official Review · Reviewer_B5Gh · 2023-07-25

**Soundness:** 4 excellent
**Presentation:** 3 good
**Contribution:** 3 good
**Rating:** 7
**Confidence:** 4

**Summary:**

This paper addresses exploration via distributional RL which has, to date, been most recently performed under an _optimism in the face of uncertainty_ (OFU) paradigm. This paper loosely characterizes this OFU approach as problematic as it results in biased exploration, resulting in sub-optimal value distributions. To remedy this pitfall, this paper proposes a stochastic perturbation of the distributional Bellman operator. Clear theoretical development of the proposed operator gives way to practical algorithm development building on top of quantile regression. Empirically, the proposed algorithm PQR is demonstrated to have clear performance benefits.

**Strengths:**

The paper sets up a really interesting juxaposition between the advantages of the distRL paradigm in accounting for aleatoric uncertainty but attempting to resolve epistemic uncertainty through exploration. As set out in the introduction of the paper, this presents an appealing motivation to perhaps determine more effective exploration strategies that are cognizant of this apparent mismatch.

The methods presented in this paper to address the one-sided tendency on risk, resulting in a practical algorithm, are reasonable and seem to neatly satisfy the intended contributions set out in the paper.

I was impressed at the extent by which the authors provide sufficient background in distributional RL in Section 2. Many papers take shortcuts here however this paper lays out relevant concepts from which the remaining development of the methods introduced in the paper are easier to understand from this grounding.

The technical formulation of the PDBOO is clear and well written. The definitions help to outline important concepts that motivate the further development of the methods.

I appreciate how the authors approach Section 4, clearly laying out the objectives of their empirical study. This helps to frame the impressive experimental performance achieved by the introduced PQR algorithm.


**Weaknesses:**

It’s not clear what is meant by “without losing the risk-neutral objective” (mentioned in the abstract)…

In line 43, it could be helpful to again list out what the two types of uncertainty are to more directly connect this sentence to what has been introduced conceptually earlier in the paper.

The authors seem to use “deep” RL interchangably with “distributional” RL in the introduction. This makes the framing of the work a little challenging to follow initially. Better clarification of prior works in the non-distributional deep RL from distributional deep RL would help make this easier to understand and would strengthen the beginning sections of the paper.

The statements made in lines 50-53 are pretty strong, as in they seem to be a result of some analysis or in the worst case are derived from opinionated intuition. Some additional justification with formal explanation (whether through a toy example or reference to later section or citation to a paper) would help make the writing easier to accept and be persuaded by. This is especially important because the “one-sided tendency on risk” appears to be a major foundation for the proposed PDBOO.

It would be helpful if the authors expanded more on their discussion at the end of Section 2.2 since DLTV appears to introduce biased exploration that is addressed in this work. A couple of sentences addressing the limitations of DLTV and how the work presented in this paper addresses those would be great. The sentence that ends the preceding paragraph (Line 100-101) is a decent example of setting up a clear understanding of the contributions set forth in the paper.

Many of the claims about why PQR performs well on n-chain and in Atari are speculative. The paper would be significantly improved if more rigorous analysis connecting the theoretical treatise in Sections 3.2 and 3.3 to empirical performance and how it differs from the baselines would be great.

The discussion around the empirical results is somewhat disappointing. It would additionally be interesting to see any ablations that could be done on PQR to help tease apart the effect of the different components of the algorithm have on performance. I suppose that this is is partially addressed with the hyperparameter plots in Figure 6 but I think it would be helpful to see how sensitive the resulting policy is to the scope of the ambiguity sets used.


**Questions:**

Is the guarantee of risk-neutrality found in expectation? E.g. is it a response to evenly choosing a risk-averse obj vs a risk-seeking obj?

The writing in lines 41-45 is pretty difficult to follow. It’s apparent that the authors are taking care to concisely convey their thoughts but fail to do so clearly.

Line 46: It’s not clear what “side-effect” is being referred to here. Additionally, it would be helpful to perhaps directly reference what section in the paper is being referred to in this sentence as the authors indicate that they’ve explored the effects of OFU in deep RL and the effects it has on handling the two types of uncertainty.

Line 61: Could you call PQR a “stochastic optimization” approach to help situate the work among other prior work? This isn’t a major concern of mine. I’m just thinking that it could be an easier way to describe the approach rather than calling it a “randomized” approach.

Line 93: The loss function here is simply the Huber loss, no? It’s not overly important to include this much detail to be honest unless the paper heavily develops further insight directly from these equations (applies to Section 2 in it’s entirety). Otherwise this just feels like a regurgitation of information that’s already established in the prior literature.

Line 128-129: Why is it important to have an interpretation of different risk levels as constructing an options framework? How does this help build the proposed methods introduced in this paper?

Line 178: Does “the standard” here mean the standard distributional Bellman operator?

Line 186: What is $B_\xi$? How does this relate to the symmetric Dirichlet distribution used to construct each ambiguity set? This seems to be addressed in Lines 210-212.

**Limitations:**

Several technical statements that crucial to the dvelopment of the proposed methods/contributions are vague and are not precisely justified. For example, it would be really helpful (and would help me agree/accept the claims) in Lines 170-174 if it were shown precisely how Eqt 1 maintains risk-neutral policy performance. As currently written we have to take the authors’ word. Additionally, the statement of using a min-expectation vs. a min-max operator should be developed and justified. This is a new concept that hasn’t been discussed previously.

It’s not clear how the time-varying amiguity sets are decoupled from a changing estimate of the return distribution Z. As formulated, PDBOO seems to ignore that $Z_\theta$ is changing through time.

The paper omits at least one relevant paper, Moskovitz, et al (NeurIPS 2021) where an ensemble of distributional crtiics are used to balance between optimism and pessmism for continous control. It would be informative to hear from the authors why this paper, and its corresponding approach, should or shouldn’t be included as a comparative baseline.

>Moskovitz, Ted, et al. "Tactical optimism and pessimism for deep reinforcement learning." Advances in Neural Information Processing Systems 34 (2021): 12849-12863.

---

> ### Author Rebuttal · Authors · 2023-08-09
>
> First of all, we appreciate the time and effort you put into our paper. We will revise our paper based on all feedback to improve the quality of the paper. We have responded to the key comments below.
>
> ## Clarity of “Without losing the risk-neutral objective”.
>
> Previous work has focused on learning variations of existing objective functions such as maximum entropy or risk-sensitive. To emphasize that our work does not modify the objective of the existing distributional Bellman update, we refer to it as a risk-neutral objective. However, this does not seem to convey the meaning well, so we will change “without losing the risk-neutral objective” to “to avoid one-sided risk tendency” for clarity.
>
> ## Strong statements in line 50-53, unclear what ‘side-effect’ is being referred to here.
> Thanks for pointing this out for readability. We will revise the paper as follows:
> In Section 4, although DLTV is the first attempt to introduce OFU in distRL, we found that consistent optimism on the uncertainty of the estimated distribution still leads to biased exploration. We will refer to this side-effect as "one-sided tendency on risk", where selecting an action based on a fixed risk criterion degrades learning performance.
>
> ## Connecting the theory to empirical performance and further ablations.
> While DLTV and p-DLTV do not satisfy our sufficient conditions, **we additionally experiment with different forms of $\Delta_t$ scheduling for PQR in global response**, such as $1/t^0$ and $\sqrt{ \log t /t}$, for a more robust comparison. We will also provide a table that shows the performance of the resulting policy as the hyperparameter varies.
>
> ## Guarantee of risk-neutrality
> Risk neutrality is guaranteed by a sufficient condition on $\Delta_t$ and updating the target with an unperturbed return of selected action. For example, DLTV is a well-motivated distributional RL method with the unperturbed return updates, but it does not theoretically guarantee the risk neutrality because the bonus decay schedule is heuristically defined by the Hoeffding inequality.
> Empirically, selecting behaviors based on randomized risk criteria plays an important role in eliminating the biased exploration of traditional OFUs, where behaviors with high intrinsic uncertainty are explored only because of the consistent positive bonuses (we call this "one-sided tendency on risk"). In our experiments, we confirm the above claim by showing that p-DLTV with only Gaussian noise injection outperforms DLTV with fixed optimism.
>
> ## Interpreting as a stochastic optimization.
> There is a slight difference. Stochastic optimization aims to obtain a robust solution from a fixed ambiguity set, while we shrink the ambiguity set to avoid one-sided risk tendency, so our goal is different. The word "randomized" is a common expression in the exploration literature [1,2], and we think it is an appropriate description because our approach is aligned with the existing works. We will explain these differences in more detail in lines 170-179 of the revised version.
>
> [1] Haque Ishfaq, Qiwen Cui, Viet Nguyen, Alex Ayoub, Zhuoran Yang, Zhaoran Wang, Doina Precup, and Lin Yang. Randomized exploration in reinforcement learning with general value function approximation. In International Conference on Machine Learning, pages 4607–4616. PMLR, 2021.
>
> [2] Ian Osband, Benjamin Van Roy, Daniel J Russo, Zheng Wen, et al. Deep exploration via randomized value functions. J. Mach. Learn. Res., 20(124):1–62, 2019.
>
> ## Relation with $B_{\xi}$ and the construction of ambiguity set
>  $B_{\xi}$ is a constant that describes uniform boundedness (such as $L$-Lipschitz, smoothness) and we just use it for theoretical rigor. Since we consider $\xi$ to be the weight for $N$ quantiles, $B_{\xi}$ is always less than or equal to $N$. It plays no direct role in constructing the ambiguity set. To avoid confusion, we will use this expression only in the appendix for the proof and remove it from the main text.
>
> ## How does Eq1 maintain risk-neutral policy performance? More discussion on min-expectation vs min-max operator.
> If we use a min-max operator with respect to risk, the solution is always more conservative (pessimistic) than the standard solution as it always considers the worst-case scenario. On the other hand, we considered the average-case which has the possibility to maintain risk-neutrality. Specifically, we aimed to find a sufficient condition to ensure risk-neutrality by scheduling the ambiguity set of risks. As such, we only share the definition of ambiguity set by the DRO literature, but with a different goal of risk-neutrality.
> PDBOO seems to ignore that $Z_\theta$ is changing through time.
> Line 156 defines PDBOO for a fixed xi, but we extend it for a time-varying PDBOO below, denoted $\xi_t$. The ambiguity set is to sample a perturbation $\xi$ that yields the same weighted expectation of the return distribution $Z$ for all actions.
>
> [Moskovitz, et al.] as a comparative baseline.
> Thanks for suggesting a novel related work. As I understand it, their work controls pessimism/optimism based on the Exponentially Weighted Average Forecasting algorithm in the MAB literature. In this manner, this work has a similar point in terms of changing the risk criteria but it is questionable whether the EWAF algorithm is still valid in deep RL where the agent updates the target and uses a function approximation.
> It is difficult to use TOP-TD3 as a baseline because our work is based on a value-based algorithm targeting a discrete action space, while TOP-TD3 experimented with continuous control using an actor-critical architecture. Note that our main baselines QR-DQN, DLTV, IQN, and RAINBOW were all experimented on the Atari environment, which is a discrete action space. However, since TOP-TD3 has a similar goal to address intrinsic(aleatoric) uncertainty in exploration, we will add it to our related work.

---

> > ### Comment · Reviewer_B5Gh · 2023-08-13
> > **Thank you for the detailed responses**
> >
> > I appreciate the detailed responses from the authors. I am satisfied by the answers which have addressed my major concerns and correcting some areas that I misunderstood. As a result, I have decided to raise my score from 6 to 7.
> >
> > Based on the responses and demonstrated efforts to revise the paper (as recommended by all reviewers) I am confident that the promised changes will result in a stronger paper, one suitable for publication.

---

> > > ### Author Response · Authors · 2023-08-14
> > > **Thanks for your positive response**
> > >
> > > We're glad that our answer satisfied you, and we appreciate your positive feedback.
> > > We'll incorporate your constructive feedback into the revised version.

---

### Author Rebuttal · Authors · 2023-08-09

# Global Response from Author

We thank all the reviewers for their valuable comments on our paper.

First, we added a revised **description of the pseudocode** and **additional experimental results for the schedule of $\Delta_t$** in the PDF, based on feedback from some reviewers.

The additional experiments were conducted in Pong-v4 where the maximum score is 21.0 and the baseline is described as follows:

- PQR : Our main algorithm, $\Delta_t$ = O(1/t^{1+\epsilon} )

- $\frac{1}{t^0}$ : $\Delta_t = O(1)$ to check the results for a fixed size ambiguity set, and it can be seen that it affects the convergence.

- $\sqrt{ \frac{\log t}{t}} $: $\Delta_t = O(\sqrt{\log t/t})$ as in the scheduling of DLTV, which theoretically does not correspond to the sufficient condition we presented.

- OPT : The output vector sampled from the Dirichlet distribution is fixed to $[0,0,...,1]$, forcing the agent to estimate only optimistically.

- $\sqrt{ \frac{\log t}{t}} $ + OPT : A hybrid of the two methods above.

Pong-v4 is a simple and easy environment where a maximum score of 21.0 can easily be achieved for many distRL algorithms(QR-DQN, IQN, RAINBOW).
The game's description for Pong-v4 is as follows:

*A player or team scores one point when the opponent hits the ball out of bounds or misses a hit.  The first player or team to score 21 points wins the game.*

In this experiment, our proposed PQR is the only method that stably achieves the maximum score with very low variance.
In the case of optimism, we can see that it performs quickly in the early stages of learning, but converges without reaching the maximum score, which is similar to the behavior of N-Chain.
In the case of fixed ambiguity, it converges to suboptimal and shows very low performance, showing the necessity of a time-varying schedule. Finally, when mimicking a schedule of p-DLTV is applied to PQR, the performance also degrades. From this, we believe that the proposed sufficient condition is quite tight.





Also, We would like to summarize and answer some common issues raised by several reviewers.

## Definition of a risk-neutral objective
Several RL papers propose new frameworks with different roles of optimal solutions by modifying the original objective functions. (e.g., maximum entropy RL and risk-sensitive RL.) In this paper, we propose the novel Bellman operator that **even if the objective function is modified, the optimal solution remains the same as before**, but only the exploration performance is improved. Since we want to obtain the same optimal solution as the original, even if we change the objective, we call it a *risk-neutral objective*, which tries to avoid one-sided risk tendency.

## Why the randomized approach is more effective than OFU in deep RL
While OFU is a principled criterion for exploration in bandit or tabular RL, it is already known that randomized approaches, including Thompson sampling, empirically outperform OFU, while there are not many papers that theoretically analyze the exact reason why.

Meanwhile, in deep RL, there have been attempts to use the information for exploration from distRL, which estimates the return distribution. DistRL tries to capture intrinsic (aleatoric) uncertainty, but the variance of the estimated distribution is mixed with parametric (epistemic) uncertainty.

We argue that using the estimated variance as the OFU is problematic and can be mitigated by a randomized approach.
The reason is that **the intrinsic uncertainty that remains in the estimated variance is irreducible during learning, and applying the OFU will bias the exploration towards risk-seeking behaviors.** We demonstrate this biased exploration phenomenon and its significant performance degradation. We also show in numerous experimental settings that the randomized approach is highly effective in avoiding this bias. Furthermore, we provide a sufficient condition for PDBOO to converge to the original optimal solution while applying a randomized risk criterion.

---

### Decision · Program_Chairs · 2023-09-21

**Decision:**

Accept (poster)

**Comment:**

This paper contributes a method for exploration in RL using distributional reinforcement learning. Initial reviews for this paper were mixed. Several reviewers found the paper clear, with good background on relevant topics within distributional RL given. Many reviewers also found the contributions novel, and the theoretical analysis solid. Some concerns mentioned in the initial reviews included the strength of the main theoretical results (Reviewer rSdX), and the potential for further explanation/examination of the empirical results obtained in the Atari suite of environments.

After detailed discussion with the authors, several reviewers have updated their scores and we now have the majority of reviewers recommending acceptance. Reviewer jYGR's recommendation of "borderline reject" remains, although we have not heard from this reviewer since the initial review. Their review raises several specific questions, such as how principled the perturbations of the DBO are, and the nature of the uncertainty sets considered. They also ask several more minor questions. In my view, the authors address these queries satisfactorily in their rebuttal. The authors also addressed several more minor concerns raised regarding the theory in their rebuttal. The recommendation is therefore to accept the paper. I would encourage the authors to take the feedback of the reviewers into account when preparing the camera-ready version.